# Rethinking and Scaling Up Graph Contrastive Learning: An Extremely Efficient Approach with Group Discrimination

**Yizhen Zheng**[1], **Shirui Pan**[2]*, **Vincent CS Lee**[1], **Yu Zheng**[3], **Phillip S. Yu**[4],

[1]Monash University, [2]Griffith University, [3]La Trobe University, [4] University of Illinons at Chicago
yizhen.zheng1@monash.edu, s.pan@griffth.edu.au, vincent.cs.lee@monash.edu
yu.zheng@latrobe.edu.au, psyu@uic.edu

## Abstract

Graph contrastive learning (GCL) alleviates the heavy reliance on label information for graph representation learning (GRL) via self-supervised learning schemes. The core idea is to learn by maximising mutual information for similar instances, which requires similarity computation between two node instances. However, GCL is inefficient in both time and memory consumption. In addition, GCL normally requires a large number of training epochs to be well-trained on large-scale datasets. Inspired by an observation of a technical defect (i.e., inappropriate usage of Sigmoid function) commonly used in two representative GCL works, DGI and MVGRL, we revisit GCL and introduce a new learning paradigm for self-supervised graph representation learning, namely, Group Discrimination (GD), and propose a novel GD-based method called G̲raph G̲roup D̲iscrimination (GGD). Instead of similarity computation, GGD directly discriminates two groups of node samples with a very simple binary cross-entropy loss. In addition, GGD requires much fewer training epochs to obtain competitive performance compared with GCL methods on large-scale datasets. These two advantages endow GGD with very efficient property. Extensive experiments show that GGD outperforms state-of-the-art self-supervised methods on *eight* datasets. In particular, GGD can be trained in 0.18 seconds (6.44 seconds including data preprocessing) on ogbn-arxiv, which is **orders of magnitude (10,000+) faster than GCL baselines** while consuming much less memory. Trained with 9 hours on ogbn-papers100M with billion edges, GGD outperforms its GCL counterparts in both accuracy and efficiency.

## 1 Introduction

Graph Neural Networks (GNNs) have been widely-adopted in learning representations for graph-structured data. By utilising message-passing over the topology of a graph, GNNs can learn effective low-dimensional node embeddings, which can be used for a variety of downstream tasks such as node classification [1]. GNNs have been further applied in diverse domains, e.g., federated learning [2, 3], trustworthy systems [4, 5], dynamic graphs [6, 7] and anomaly detection [8, 9].

However, many GNNs adopt a supervised learning manner to train models with label information, which is expensive and labour-intensive to collect in real-world. To address this issue, a few studies (e.g., DGI [10], MVGRL [11], GMI [12], and GRACE [13]) borrow the idea of contrastive learning from computer vision (CV), and introduce graph contrastive learning (GCL) methods for self-supervised GRL. The core idea of these methods is to maximise the mutual information (MI) between an anchor node and its positive counterparts, sharing similar semantic information while doing the

---

*Corresponding Author.

opposite for negative counterparts as shown in Figure 1(a). Nonetheless, such a scheme relies on similarity calculation in contrastive loss computation. Additionally, GCL normally requires a large number of training epochs to be well-trained on large-scale datasets. Thus, when the size of the dataset is large, these methods require a significant amount of time and resources to be well-trained.

Though a few GCL works attempt to improve graph contrastive learning with specially designed schemes, e.g., BGRL [15] and GBT [14], they are still inefficient and require high time consumption for model training. Inspired by BYOL [16], BGRL [15] adopts a bootstrapping scheme and remove negative node pairs. It only contrasts a node from the online network (i.e., updated with gradient) to its corresponding embedding from the target network (i.e., updated momentumly with stop gradient). Based on Barlow-Twins [17],

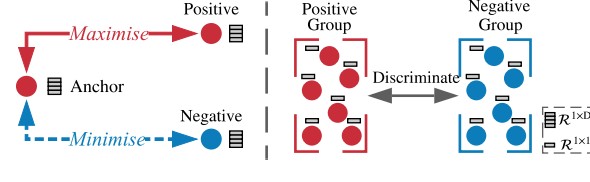

(a)Graph Contrastive Learning  (b) Group Discrimination

Figure 1: The left subfigure shows the GCL learning scheme. Red line indicates MI maximisation between two nodes, each of which $\in \mathbb{R}^{1 \times D}$, while blue line indicates the opposite operation. The right subfigure presents Group Discrimination. It discriminates positive and negative node samples, each of which $\in \mathbb{R}^{1 \times 1}$.

GBT [14] borrows the idea of redundancy-reduction principle and utilises a cross-correlation-based loss to build contrastiveness between embedding dimensions.

To boost training efficiency of self-supervised GRL, inspired by an observation of a technical defect (i.e., inappropriate application of Sigmoid function) in two representative GCL studies, we introduce a novel learning paradigm, namely, Group Discrimination (GD). Instead of similarity computation, GD directly discriminates a group of positive nodes from a group of negative nodes, as shown in Figure 1(b). Specifically, GD defines node samples generated with original graph as the positive group, while node samples obtained with corrupted topology are regarded as the negative group. Then, GD trains the model by classifying these node samples into the correct group with a very simple binary cross-entropy loss. By doing so, the model can extract valuable self-supervised signals from learning the edge distribution of a graph. Com-

Table 1: Training time in seconds comparison between GGD and GBT [14] (i.e., the most efficient GCL baseline as shown in section 5.1) on ogbn-arxiv. Number in brackets means the hidden size. 'Pre', 'Tr' and 'Epo' indicate preprocessing time, training time per epoch, and the number of epochs for training GNNs. 'Total(E)' and 'Total(T)' are total end-to-end training time (i.e., including pre-processing), which equals to $(\mathrm{Pre} + \mathrm{Epo} \times \mathrm{Tr})$ and total training time, which is $(\mathrm{Epo} \times \mathrm{Tr})$. 'Imp(E)' and 'Imp(T)' indicate how many times GGD improve on 'Total(E)' and 'Total(T)'. 'Acc' is averaged accuracy result on test set over five runs.

| Method | Pre | Tr | Epo | Total(E) | Imp(E) | Total(T) | Imp(T) | Acc |
|--------|-----|-----|-----|----------|--------|----------|--------|-----|
| GBT(256) | 5.52 | 6.47 | 300 | 1,946.52 | - | 1,941.00 | - | 70.1 |
| GGD(256) | 6.26 | 0.18 | 1 | 6.44 | 302.25 × | 0.18 | 10,783.33× | 70.3 |
| GGD(1,500) | 6.26 | 0.95 | 1 | 7.21 | 269.96× | 0.95 | 2,043.16× | 71.6 |

pared with GCL, GD enjoys numerous merits including extremely fast training, fast convergence (e.g., 1 epoch to be well-trained on large-scale datasets), and high scalability while achieving SOTA performance with existing GCL approaches.

Using GD as backbone, we design a new self-supervised GRL model with the Siamese structure called Graph Group Discrimination (GGD). Firstly, we can optionally augment a given graph with augmentation techniques, e.g., feature and edge dropout. Then, the augmented graph is fed into a GNN encoder and a projector to obtain embeddings for the positive group. After that, the augmented feature is corrupted with node shuffling (i.e., disarranging the order of nodes in the feature matrix) to disrupt the topology of a graph and input to the same network for obtaining embeddings of the opposing group. Finally, the model is trained by discriminating these two groups of node samples. The contributions of this paper are three-fold: 1) We re-examine existing GCL approaches (e.g., DGI [10] and MVGRL [11]), and we introduce a novel and efficient self-supervised GRL paradigm, namely, Group Discrimination (GD). 2) Based on GD, we propose a new self-supervised GRL model, GGD, which is fast in training and convergence, and possess high scalability. 3) We conduct extensive experiments on eight datasets, including an extremely large dataset, ogbn-papers100M with billion edges. The experiment results show that our proposed method **reaches state-of-the-art performance while consuming much less time and memory than baselines, e.g., 10783 × faster** than the most efficient GCL baseline with its best selected epochs number [14], as shown in Table 1.

## 2 Rethinking Representative GCL Methods

In this section, we analyse a technical defect observed in two representative GCL methods, DGI [10] and MVGRL [11]. Based on the technical defect, we show that mutual information maximisation behind these two approaches is not the contributed factor to contrastive learning, but a new paradigm, group discrimination. Finally, from the analysis, we provide the definition of this new concept.

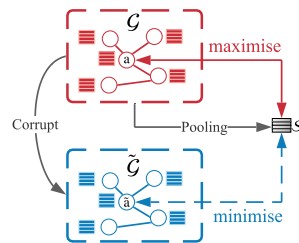

Figure 2: The architecture of DGI. Cubes indicate node embeddings. Red and blue lines represent MI maximisation and minimisation, respectively. $\mathcal{G}$ and $\tilde{\mathcal{G}}$ denote the original graph and the corrupted graph. $\mathbf{s}$ is the summary vector.

### 2.1 Rethinking GCL Methods

DGI [10] is the first work introducing contrastive learning into GRL. However, due to a technical defect observed in their official open-source code, we found it is essentially not working as the authors thought (i.e., learning via MI interaction).

**Constant Summary Vector.** As shown in Figure 2, the original idea of DGI is to maximise the MI (i.e., the red line) between a node $a$ and the summary vector $\mathbf{s}$, which is obtained by averaging all node embeddings in a graph $\mathcal{G}$. Also, to regularise the model training, DGI corrupts $\mathcal{G}$ by shuffling the node order of the input feature matrix to get $\tilde{\mathcal{G}}$. Then, generated embeddings of $\tilde{\mathcal{G}}$ serve as negative samples, which are pulled apart from the summary vector $\mathbf{s}$ via MI minimisation.

Nonetheless, in the implementation of DGI, a Sigmoid function is inappropriately applied on the summary vector generated from a GNN whose weight is initialised with Xavier initialisation. As a result, elements in the summary vector are very close to the same value. We have validated this finding on three datasets, Cora, CiteSeer and PubMed. The experiment result is shown in Table 2, which shows that summary vec-

Table 2: Summary vector statistics on three datasets with different activation functions including ReLU, LeakyReLU (i.e., LReLU shown below), PReLU, and Sigmoid.

| Activation | Statistics | Cora | CiteSeer | PubMed |
|---|---|---|---|---|
| ReLU/LReLU/PReLU | Mean | 0.50 | 0.50 | 0.50 |
| | Std | 1.3e-03 | 1.0e-04 | 4.0e-04 |
| | Range | 1.4e-03 | 8.0e-04 | 1.5e-03 |
| Sigmoid | Mean | 0.62 | 0.62 | 0.62 |
| | Std | 5.4e-05 | 2.9e-05 | 6.6e-05 |
| | Range | 3.6e-03 | 3.0e-03 | 3.2e-03 |

tors in all datasets are approximately a **constant vector** $\epsilon \boldsymbol{I}$, where $\epsilon$ is a scalar and $\boldsymbol{I}$ is an all-ones vector (i.e., $\epsilon$=0.50 with ReLU/LReLU/PReLU and $\epsilon$=0.62 with Sigmoid as non-linear activation in these datasets).

To theoretically explain this phenomenon, we present the proposition below:

**Proposition 1** *Given* $\mathcal{G} = \{\mathbf{X} \in \mathbb{R}^{N \times D}, \mathbf{A} \in \mathbb{R}^{N \times N}\}$, *and a GCN encoder* $g(\cdot)$ *initialised with Xavier initialisation, we can obtain its embedding* $\mathbf{H} = \sigma(g(\mathcal{G}))$, *where* $\sigma(\cdot)$ *is a non-linear activation function. By applying the sigmoid function* $\sigma_{sig}(\cdot)$ *to the summary vector* $\mathbf{s}$ *(i.e., the average row vector of* $\mathbf{H}$*), values in* $\sigma_{sig}(\mathbf{s})$ *approximately become 0.5 with ReLU/LReLU/PReLU or 0.62 with Sigmoid as non-linear activation of* $g(\cdot)$ *at the initialisation stage.*

Table 3: The experiment result on three datasets with changing value from 0 to 1.0 for the summary vector.

| Dataset | 0 | 0.2 | 0.4 | 0.6 | 0.8 | 1.0 |
|---|---|---|---|---|---|---|
| Cora | 70.3±0.7 | 82.4±0.2 | 82.3±0.3 | 82.5±0.4 | 82.3±0.3 | 82.5±0.1 |
| CiteSeer | 61.8±0.8 | 71.7±0.6 | 71.9±0.7 | 71.6±0.9 | 71.7±1.0 | 71.6±0.8 |
| PubMed | 68.3±1.5 | 77.8±0.5 | 77.9±0.8 | 77.7±0.9 | 77.4±1.1 | 77.2±0.9 |

Based on this proposition, we can see these summary vectors can lose variance and become a constant vector at the initialisation stage. Based on Table 2, we can see the constant in the summary vector remain unchanged, and the information loss still occurs even if the GNN encoder is trained. Thus, we conjecture the training process won't affect the constant value much in the summary vector of DGI. The proof for the proposition is presented in Appendix A.1.

To evaluate the effect of $\epsilon$ to constant summary vector, we vary the scalar $\epsilon$ (from 0 to 1 increment by 0.2) to change the constant summary vector and report the model performance (i.e., averaged accuracy on five runs) in Table 3.

From this table, we can see, except for 0, the model performance is trivially affected by $\epsilon$ for constant summary vector. When the summary vector is set to 0, the model performance plummets because node embeddings become all 0 when multiplying with such vector and the model converges to the trivial solution. As the summary vector only has a trivial effect on model training, the hypothesis of DGI [10] on learning via contrastiveness between anchor nodes and the summary instance does not hold, which raises a question to be investigated: *What truly leads to the success of DGI?*

**Simplifying DGI.** To answer the question, we predigest the objective function proposed in DGI (i.e., maximising the MI between $\mathbf{h}_i$ and the summary vector $\mathbf{s}$) by using an all-ones vector as the summary vector $\mathbf{s}$ (i.e., setting $\mathbf{s} = \epsilon \boldsymbol{I} = \boldsymbol{I}$) and simplifying the discriminator $\mathcal{D}(\cdot)$ (i.e., removing the learnable weight matrix). Then, we rewrite the objective function to the following form:

$$
\begin{aligned}
\mathcal{L}_{DGI} &= \frac{1}{2N}(\sum_{i=1}^{N} \log \mathcal{D}(\mathbf{h}_i, \mathbf{s}) + \log(1 - \mathcal{D}(\tilde{\mathbf{h}}_i, \mathbf{s}))), \\
&= \frac{1}{2N}(\sum_{i=1}^{N} \log(\mathbf{h}_i \cdot \mathbf{s}) + \log(1 - \tilde{\mathbf{h}}_i \cdot \mathbf{s})), \\
&= \frac{1}{2N}(\sum_{i=1}^{N} \log(sum(\mathbf{h}_i)) + \log(1 - sum(\tilde{\mathbf{h}}_i))),
\end{aligned}
\tag{1}
$$

where $\cdot$ is the vector multiplication operation, $N$ is the number of nodes in a graph, $\mathbf{h}_i \in \mathbb{R}^{1 \times D}$ and $\tilde{\mathbf{h}}_i \in \mathbb{R}^{1 \times D}$ are the original and corrupted embedding for node $i$, $sum(\cdot)$ is the summation function, and $\mathcal{D}(\cdot)$ is a discriminator for bilinear transformation, which can be formulated as follows:

$$
\mathcal{D}(\mathbf{h}_i, \mathbf{s}) = \sigma_{sig}(\mathbf{h}_i \cdot \mathbf{W} \cdot \mathbf{s}),
\tag{2}
$$

where $\mathbf{W}$ is a learnable weight matrix and $\sigma_{sig}(\cdot)$ is the sigmoid function. Specifically, as shown in Equation 2, by removing the weight matrix $\mathbf{W}$, $\mathbf{h}_i$ is directly multiplied with $\mathbf{s}$. As $\mathbf{s}$ is a vector containing only one, the multiplication of $\mathbf{h}_i$ and $\mathbf{s}$ is equivalent to summing $\mathbf{h}_i$ itself directly. From this form, we can see that the multiplication of $\mathbf{h}_i$ and the summary vector only serves as an aggregation function (i.e., summation aggregation) to aggregate $\mathbf{h}_i$. To explore the effect of other aggregation functions, we replace the summation function in Equation 1 with other aggregation methods such as mean-, minimum-, and maximum- pooling, and present the experiment result in Appendix A.3.

Table 4: Comparison of the original DGI and $\text{DGI}_{BCE}$ in terms of accuracy (averaged on five runs), memory efficiency (in MB) and training time (in seconds). Number after | shows how many times have $\text{DGI}_{BCE}$ improved on top of DGI.

| Experiment | Method | Cora | CiteSeer | PubMed |
|---|---|---|---|---|
| Accuracy | DGI | 81.7±0.6 | 71.5±0.7 | 77.3±0.6 |
| | $\text{DGI}_{BCE}$ | 82.5±0.3 | 71.7±0.6 | 77.7±0.5 |
| Memory | DGI | 4189MB | 8199MB | 11471MB |
| | $\text{DGI}_{BCE}$ | 1475MB\|64.8% | 1587MB\|80.6% | 1629MB\|85.8% |
| Time | DGI | 0.085s | 0.134s | 0.158s |
| | $\text{DGI}_{BCE}$ | 0.010s\|8.5× | 0.021s\|6.4× | 0.015s\|10.5× |

Based on Equation 1, we can rewrite it to a very simple binary cross entropy loss if we also include corrupted nodes as data samples and setting $\hat{y}_i = agg(\mathbf{h}_i)$, where $agg(\cdot)$ stands for aggregation:

$$
\mathcal{L}_{BCE} = -\frac{1}{2N}(\sum_{i=1}^{2N} y_i \log \hat{y}_i + (1 - y_i) \log(1 - \hat{y}_i)),
\tag{3}
$$

where $y_i \in \mathbb{R}^{1 \times 1}$ means the indicator for node $i$ (i.e., if node $i$ is corrupted, $y_i$ is 0, otherwise it is 1), and $\hat{y}_i \in \mathbb{R}^{1 \times 1}$ represents the prediction for a node sample $i$. As we include corrupted nodes as data samples, the size of nodes to be processed is doubled to $2N$ (i.e., the number of corrupted nodes is equal to the number of original nodes). From the equation above, we can easily observe that what DGI truly does is discriminate between a group of nodes generated with correct topology and nodes generated with corrupted topology, as shown in Figure 1. We name this self-supervised learning paradigm "**Group Discrimination**". To validate the effectiveness of this paradigm, we replace the original DGI loss with Equation 3, namely, $\text{DGI}_{BCE}$ and compare it with DGI on three datasets in terms of training time, memory efficiency and model performance as shown in Table 4. Here, $\text{DGI}_{BCE}$ adopts the same parameter setting as DGI. From this table, we can observe $\text{DGI}_{BCE}$ dramatically improves DGI in both memory and time efficiency while it slightly enhances the model performance of DGI. This may be contributed to the removal of multiplication operations between node pairs, which eases the burden of computation and memory consumption.

Similar to DGI, the same technical defect is observed in MVGRL [11], which makes it become a GD-based method. Extended on DGI, MVGRL [11] incorporates diffusion augmentation to inject additional global information into model training, which enhances the model performance. The detailed analysis for MVGRL is presented in Appendix A.4.

## 2.2 Definition of Group Discrimination

As mentioned above, Group Discrimination is a self-supervised GRL paradigm, which learns by discriminating different groups of node samples. Specifically, the paradigm assigns different indicators to different groups of node samples. For example, for binary group discrimination, one group is considered as the positive group with class 1 as its indicator, whereas the other group is the negative group, having its indicator assigned as 0. Given a graph $\mathcal{G}$, the positive group usually includes node samples generated with the original graph $\mathcal{G}$ or its augmented views (i.e., similar graph instances of $\mathcal{G}$ created by augmentation). In contrast, the opposing group contains negative samples obtained by corrupting $\mathcal{G}$, e.g., changing its topology structure.

Based on our theoretical analysis, group discrimination is learning to avoid making 'mistakes' (i.e., bias the encoder towards avoiding mistaken samples), thus improving the quality of generated embeddings. The analysis and an intuitive explanation are presented in Section 6.1.1 and A.2.1.

## 3 Methodology

We first define unsupervised node representation learning and then present the architecture of GGD, which extends $\mathbf{DGI}_{BCE}$ with additional augmentation, the projector and embedding reinforcement to reach better model performance. Given a graph $\mathcal{G}$ with attributes $\mathbf{X} \in \mathbb{R}^{N \times D}$, where $N$ is the number of nodes in $\mathcal{G}$, and $D$ is the number of dimensions of $\mathbf{X}$, our aim is to train a GNN encoder without the reliance on labelling information. With the trained encoder, taking $\mathcal{G}$ and $\mathbf{X}$ as input, it can output learned representations $\mathbf{H} \in \mathbb{R}^{N \times D'}$, where $D'$ is the predefined hidden dimension. $\mathbf{H}$ can then be used in many downstream tasks such as node classification.

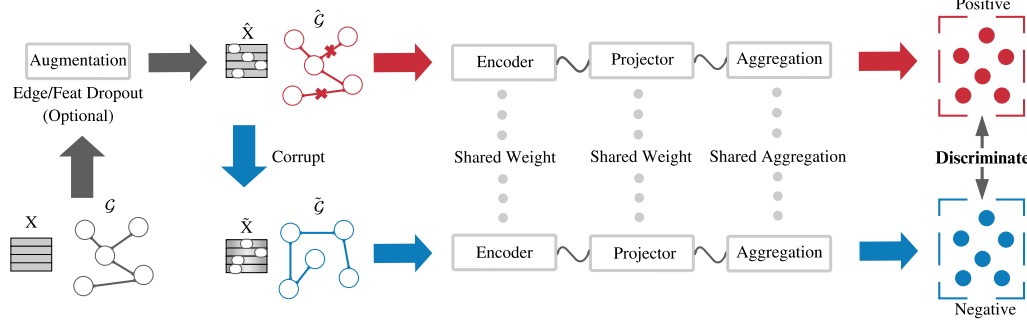

Figure 3: The architecture of GGD. Given a graph $\mathcal{G}$ with a feature matrix $\mathbf{X}$, we can optionally apply augmentation on them to generate $\hat{\mathcal{G}}$ and $\hat{\mathbf{X}}$. Then, we corrupt $\hat{\mathbf{X}}$ and $\hat{\mathcal{G}}$ to obtain $\tilde{\mathbf{X}}$ and $\tilde{\mathcal{G}}$. Taking $\hat{\mathbf{X}}$ and $\hat{\mathcal{G}}$ as input to the encoder and the projector, i.e., a multilayer perceptron, positive node samples can be obtained. Similarly, $\tilde{\mathbf{X}}$ and $\tilde{\mathcal{G}}$ are fed to the same encoder and projector to generate negative samples. The generated embeddings are aggregated to get predictions for the group discrimination task. This process will be iteratively conducted until reaching the predefined training epochs.

### 3.1 Graph Group Discrimination

Based on the proposed self-supervised GRL paradigm, group discrimination, we have designed a novel method, namely GGD, to learn node representations using a siamese network structure and a BCE loss. The architecture of GGD is presented in Figure 3. The framework mainly consists of four components: augmentation, corruption, a siamese GNN network, and group discrimination.

**Augmentation.** With a given graph $\mathcal{G}$ and feature matrix $\mathbf{X}$, optionally, we can augment it with augmentation techniques such as edge and feature dropout to create $\hat{\mathcal{G}}$ and $\hat{\mathbf{X}}$. In practice, we follow the

augmentation proposed in GraphCL [18]. Specifically, edge dropout removes a predefined fraction of edges, while we use node dropout to mask a predefined proportion of feature dimension, i.e., assigning 0 to replace values in randomly selected dimensions. This step is optional in implementation.

Notably, the motivation of using augmentation in our framework is distinct from contrastive learning methods. In our study, augmentation is used to increase the difficulty of the self-supervised training tasks. With augmentation, $\hat{\mathcal{G}}$ and $\hat{\mathbf{X}}$ change in every training iteration, which forces the model to lessen the dependence on the fixed pattern (i.e., unchanged edge and feature distribution) in a monotonous graph. However, in contrastive learning, augmentation creates augmented views sharing similar semantic information for building contrastiveness.

**Corruption.** $\hat{\mathcal{G}}$ and $\hat{\mathbf{X}}$ are then corrupted to build $\tilde{\mathcal{G}}$ and $\tilde{\mathbf{X}}$ for the generation of node embeddings in the negative group. We adopt the same corruption technique used in DGI [10] and MVGRL [11] (as shown in Figure 5). The corruption technique devastates the topology structure of $\hat{\mathcal{G}}$ by randomly changing the order of nodes in $\hat{\mathbf{X}}$. The corrupted $\tilde{\mathbf{X}}$ and $\tilde{\mathcal{G}}$ can be used for producing node representations with incorrect network connections.

**The Siamese GNN.** We have designed a siamese GNN network to output node representations given a graph and its attribute. The siamese GNN network is made up of two components, which are a GNN encoder and a projector. The backbone GNN encoder is replaceable with a variety of choices of GNNs, e.g., GCN [19] and GAT [20]. In our work, we adopt GCN as the backbone. The projector is a multi-layer perceptron network, whose number of layers can be adjusted. When generating node embeddings of the positive group, the Siamese network takes $\hat{\mathcal{G}}$ and $\hat{\mathbf{X}}$ as input. Using the same encoder and projector, the Siamese network output the negative group with $\tilde{\mathcal{G}}$ and $\tilde{\mathbf{X}}$. These two groups of node embeddings are considered as a collection of data samples with a size of $2N$ for discrimination. Before conducting group discrimination, in the "aggregation" phase, all data samples are aggregated with the same aggregation technique, e.g., sum-, mean-, and linear aggregation.

**Group Discrimination.** In the group discrimination process, we adopt a very simple binary cross entropy (BCE) loss to discriminate two groups of node samples as shown in Equation 3. In our implementation, $y_i$ is 0 and 1 for node embeddings in negative and positive groups. During model training, the model is optimised by categorising node embeddings in the collection of data samples into their corresponding class correctly. The loss is computed by comparing the prediction of a node $i$, i.e., a scalar, with its indicator $y_i$. With the ease of BCE loss computation, the training process of GGD is very fast and memory efficient.

### 3.2 Model Inference

During training, the model is optimised via loss minimisation with Equation 3. The time complexity analysis of GGD is provided in Appendix A.5. In the inference phase, we freeze the trained GNN encoder $g_\theta$ and obtain node embeddings $\mathbf{H}_\theta$ with the input $\mathcal{G}$.

Inspired by MVGRL [11], which strengthens the output embeddings by including additional global information, we adopt a conceptually similar embedding reinforcement approach. Specifically, they obtain the final embeddings by summing up embeddings from two views: the original view comprising local information and the diffused view with global information. This operation reinforces the final embeddings and leads to model performance improvement. Nonetheless, graph diffusion impairs the scalability of a model [21] and hence cannot be directly applied in our embedding generation process. To avoid the diffusion computation, we have come up with a workaround in the virtue of the power of a graph to extract global information. The power of a graph can extend the message passing scope of $\mathbf{H}_\theta$ to n-hop neighbourhood, which encodes global information from distant neighbours. It can be formulated as follows:

$$\mathbf{H}_\theta^{global} = \mathbf{A}^n \mathbf{H}_\theta, \tag{4}$$

where $\mathbf{H}_\theta^{global}$ is the global embedding, and $\mathbf{A}$ is the adjacency matrix of the graph $\mathcal{G}$. It is notable that this operation can be easily decomposed with the associative property of matrix multiplication and is easy to compute. To show the easiness of such computation, we conduct an experiment showing its time consumption on various datasets in Appendix A.6. Finally, the final embedding can be achieved by $\mathbf{H} = \mathbf{H}_\theta^{global} + \mathbf{H}_\theta$, which can be used for downstream tasks. In our experiment, we conduct node classification tasks. Following the common practice of GCL methods [10, 13, 15, 12, 17], these tasks are performed by using the final embeddings $\mathbf{H}$ to train and test a simple logistic regression classifier.

## 4 Related Work

**Graph Neural Networks (GNNs).** are generalised deep neural networks for graph-structured data. GNNs mainly have two categories, spectral-based GNNs and spatial-based GNNs. Spectral GNNs attempt to use eigen-decomposition to obtain the spectral-based representation of graphs, whereas spatial GNNs focus on using spatial neighbours of nodes for message passing. Extending spectral-based methods to the spatial domain, GCN [19] utilises first-order Chebyshev polynomial filters to approximate spectral-based graph convolution. Taking the weight of spatial neighbours in consideration, GAT [20], improves GCN by introducing attention module in message passing. To decouple message passing from neural networks, SGC [22] simplifies GCN by removing non-linearity and weight matrices in graph convolution layers. However, these studies cannot handle datasets with limited or no labels. Graph contrastive learning has been recently exploited to address this issue.

**Graph Contrastive Learning (GCL).** aims to alleviate the reliance on labelling information in model training based on the concept of mutual information (MI). Specifically, GCL approaches maximise MI between instances with similar semantic information, and minimise MI between dissimilar instances. For example, DGI [10] builds contrastiveness between node embeddings and a summary vector (i.e., a graph level embedding obtained by averaging all node embeddings) with a JSD estimator. To improve DGI, MVGRL [11] and GMI [12] extends the idea of DGI by introducing multi-view contrastiveness with diffusion augmentation, and focusing on a local scope with the first-order neighbourhood, respectively. Adopting InfoNCE loss, GRACE [13] applies augmentation techniques to create two augmented views and inject contrastiveness between them. Though these GCL methods have successfully outperformed some supervised baselines in benchmark datasets, these methods suffer from significant limitations, including time-consuming training, memory inefficiency, and poor scalability. In contrast, GGD requires much less time in training and posses high scalability.

**Scalable GNNs.** Efficiency is a bottleneck for most existing GNNs to handle large graphs. To address this challenge, there are mainly three categories of approaches: layer-wise sampling (e.g., GraphSage [23]), graph sampling methods such as Cluster-GCN [24] and GraphSAINT [25], and linear models, e.g., SGC [22] and PPRGo [26]. GraphSage [23] introduces a neighbour-sampling approach, which creates fixed-size subgraphs for each node. Underpinned by graph sampling, Cluster-GCN [24] decomposes a large-scale graph into multiple subgraphs based on clustering, while GraphSAINT [25] utilises light-weight graph samplers along with a normalisation technique for biases elimination in mini-batches. Linear models, SGC [22] and PPRGo [26], decouple graph convolution from embedding transformation (i.e., matrix multiplication with weight matrices), and leverage Personalised PageRank to encode multi-hop neighbourhood, respectively. However, all these methods only focus on supervised learning on graphs. For unsupervised/self-supervised learning settings where no labelled supervision signal is available, these frameworks are not applicable. The closest works to ours to handle large scale graph datasets under self-supervised settings are BGRL [15] and GBT [14]. They try to improve the contrastive losses by removing negative samples. However, BGRL [15] and GBT [14] still require much more time in training compared with GGD.

## 5 Experiments

We evaluate the effectiveness of our model using eight benchmark datasets of different sizes. These datasets include five small- and medium-scale datasets: Cora, CiteSeer, PubMed [27], Amazon Computers, and Amazon Photos [28], as well as large-scale datasets ogbn-arxiv, ogbn-products and ogbn-papers100M. Notably, ogbn-papers100M is the largest dataset provided by Open Graph Benchmark[29] for node property prediction tasks. It has over 110 million nodes and 1 billion edges. The statistics of these datasets are summarised in Appendix A.7. To ensure reproducibility, **the detailed experiment settings and computing infrastructure are summarised in Appendix A.8.** The source code is already open sourced[2].

Table 5: Model performance of node classification on 5 datasets. **X, A** and **Y** represent feature, adjacency matrix, and labels. Best performance for each dataset is in **bold**. Comp and Photo refer to Amazon Computers and Amazon Photos.

| Data | Method | Cora | CiteSeer | PubMed | Comp | Photo |
|------|--------|------|----------|--------|------|-------|
| **X, A, Y** | GCN | 81.5 | 70.3 | 79.0 | 76.3±0.5 | 87.3±1.0 |
| **X, A, Y** | GAT | 83.0±0.7 | 72.5±0.7 | 79.0±0.3 | 79.3±1.1 | 86.2±1.5 |
| **X, A, Y** | SGC | 81.0±0.0 | 71.9±0.1 | 78.9±0.0 | 74.4±0.1 | 86.4±0.0 |
| **X, A, Y** | CG3 | 83.4±0.7 | **73.6**±0.8 | 80.2±0.8 | 79.9±0.6 | 89.4±0.5 |
| **X, A** | DGI | 81.7±0.6 | 71.5±0.7 | 77.3±0.6 | 75.9±0.6 | 83.1±0.5 |
| **X, A** | GMI | 82.7±0.2 | 73.0±0.3 | 80.1±0.2 | 76.8±0.1 | 85.1±0.1 |
| **X, A** | MVGRL | 82.9±0.7 | 72.6±0.7 | 79.4±0.3 | 79.0±0.6 | 87.3±0.3 |
| **X, A** | GRACE | 80.0±0.4 | 71.7±0.6 | 79.5±1.1 | 71.8±0.4 | 81.8±1.0 |
| **X, A** | GraphCL | 82.5±0.2 | 72.8±0.3 | 77.5±0.2 | OOM | 79.5±0.4 |
| **X, A** | BGRL | 80.5±1.0 | 71.0±1.2 | 79.5±0.6 | 89.2±0.9 | 91.2±0.8 |
| **X, A** | GBT | 81.0±0.5 | 70.8±0.2 | 79.0±0.1 | 88.5±1.0 | 91.1±0.7 |
| **X, A** | GGD | **83.9**±0.4 | 73.0±0.6 | **81.3**±0.8 | **90.1**±0.9 | **92.5**±0.6 |

---
[2]https://github.com/zyzisastudyreallyhardguy/Graph-Group-Discrimination

## 5.1 Evaluating on Small- and Medium-scale Datasets

We compare GGD with ten baselines including four supervised GNNs (i.e., GCN [19], GAT [20], SGC [22], and CG3 [30]) and six GCL methods (i.e., DGI [10], GMI [12], MVGRL [11], GRACE [13], BGRL [15] and GBT [14]) on five small- and medium scale benchmark datasets. In the experiment, we follow the same data splits as [31] for Cora, Cite-Seer and PubMed. For Amazon Computers and Photos, we use a random split setting, which randomly allocates 10/10/80% of data to training/validation/test set, respectively. The model performance is measured using the averaged classification accuracy with five results along with standard deviations and reported in Table 5.

Table 6: Comparison of training time per epoch in underline{seconds} between six GCL-based methods and GGD on five datasets. Improve means how many times are GGD faster than baselines. '-' means the improvement range.

| Method | Cora | CiteSeer | PubMed | Comp | Photo |
|--------|------|----------|--------|------|-------|
| DGI | 0.085 | 0.134 | 0.158 | 0.171 | 0.059 |
| GMI | 0.394 | 0.497 | 2.285 | 1.297 | 0.637 |
| MVGRL | 0.123 | 0.171 | 0.488 | 0.663 | 0.468 |
| GRACE | 0.056 | 0.092 | 0.893 | 0.546 | 0.203 |
| GraphCL | 0.073 | 0.085 | 0.123 | OOM | 0.188 |
| BGRL | 0.085 | 0.094 | 0.147 | 0.337 | 0.273 |
| GBT | 0.073 | 0.072 | 0.103 | 0.492 | 0.173 |
| GGD | 0.010 | 0.021 | 0.015 | 0.016 | 0.009 |
| Improve | 7.3-39.4× | 3.4-23.7× | 6.9-152.3× | 10.7-15.3× | 19.2-70.8× |

**Accuracy.** From Table 5, we can observe that GGD generally outperforms all baselines in all datasets. The only exception is on Cite-Seer dataset, where the semi-supervised method, CG3[30], slightly outperforms GGD, which still provides the 2nd best performance. In this experiment, we use the officially released code of GraphCL [18], BGRL [15] and GBT [14] to reproduce the result, while the other results are sourced from previous studies [30, 1].

Table 7: Comparison of memory consumption in underline{MBs} of six GCL baselines and GGD on five datasets.

| Method | Cora | CiteSeer | PubMed | Comp | Photo |
|--------|------|----------|--------|------|-------|
| DGI | 4,189 | 8,199 | 11,471 | 7,991 | 4,946 |
| GMI | 4,527 | 5,467 | 14,697 | 10,655 | 5,219 |
| MVGRL | 5,381 | 5,429 | 6,619 | 6,645 | 6,645 |
| GRACE | 1,913 | 2,043 | 12,597 | 8,129 | 4,881 |
| GraphCL | 4,163 | 8,249 | 11,555 | OOM | 9,083 |
| BGRL | 1,627 | 1,749 | 2,299 | 5,069 | 3,303 |
| GBT | 1,651 | 1,799 | 2,461 | 5,037 | 2,641 |
| GGD | 1,475 | 1,587 | 1,629 | 1,787 | 1,637 |
| Improve | 10.7-72.6% | 11.8-80.6% | 27.2-85.8% | 64.5-83.2% | 38.0-75.4% |

**Efficiency and Memory Consumption.** GGD is substantially more efficient than other self-supervised baselines in time and memory consumption as shown in Table 6 and Table 7. Remarkably, GGD is **19.2 times faster** in Amazon Photos for training time per epoch, and consumes **64.5% less memory** in Amazon Computers for memory consumption than the most efficient baseline (i.e., GBT [14]). The dramatic boost of time and memory efficiency of GGD is contributed to the exclusion of similarity computation, which enables model training without multiplication of node embeddings.

## 5.2 Evaluating on Large-scale datasets

To evaluate the scalability of GGD, we choose three large-scale datasets from Open Graph Benchmark [29], which are ogbn-arxiv, ogbn-products, and ogbn-papers100M. ogbn-papers100M is the most challenging large-scale graph available in Open Graph Benchmark for node property prediction with over 1 billion edges and 110 million nodes. Extending to extremely large graphs (i.e., ogbn-products and ogbn-papers100M), we adopt a Neighbourhood Sampling strategy, which is described in Appendix A.8.

**ogbn-arxiv & ogbn-products.** For ogbn-arxiv, we compare GGD against four self-supervised baselines (i.e., DGI [10], GRACE [13], BGRL [15]and GBT [14]), whereas BGRL [15] and GBT [14] are selected to be compared for ogbn-products. In addition, we include the performance of MLP, Node2vec [32], and supervised GCN [19] sourced from [29] in Table 8 and Table 9. For memory and training time comparison, we only compare GGD with the two most efficient baselines (i.e., BGRL and GBT according to Tables 6 and 7). In ogbn-arxiv, we reproduce BGRL [15] and found it fails to process ogbn-arxiv in full batch. Thus, we only compare GGD and GBT in this dataset, which can successfully train in full-graph processing mode.

Table 8: Node classification result and efficiency comparison on ogbn-arxiv. 'epo' means epoch. 'Time' means training time per epoch (in seconds). 'Total' is total training time (Number of epochs × 'Time'). OOM indicates out-of-memory on Nvidia A40 (48GB). Number after \ means the hidden size of GGD.

| Method | Valid | Test | Memory | Time | Total |
|--------|-------|------|--------|------|-------|
| Supervised GCN | 73.0±0.2 | 71.7±0.3 | - | - | - |
| MLP | 57.7±0.4 | 55.5±0.2 | - | - | - |
| Node2vec | 71.3±0.1 | 70.1±0.1 | - | - | - |
| DGI | 71.3±0.1 | 70.3±0.2 | - | - | - |
| GRACE(10k epos) | 72.6±0.2 | 71.5±0.1 | - | - | - |
| BGRL(10k epos) | 72.5±0.1 | 71.6±0.1 | OOM (Full-graph) | / | / |
| GBT(300 epos) | 71.0±0.1 | 70.1±0.2 | 14,959MB | 6.47 | 1,941.00 |
| GGD(1 epo\1500) | 72.7±0.3 | 71.6±0.5 | 14,666MB | 0.95 | 0.95\2,043× |
| GGD(1 epo\256) | 71.0±0.2 | 70.3±0.3 | 4,513MB\69.8% | 0.18 | 0.18\10,783× |

From Table 8 and Table 9, we can see `GGD` remarkably achieves the state-of-the-art performance using only one epoch to train. As a result, `GGD` is 10,783 times faster than the most efficient baseline, i.e., GBT [14], on total training time to reach the desirable performance in ogbn-arxiv. Please note that the number of epoches in our experiment is consistent with the optimal choice

Table 9: Node classification result and efficiency comparison on ogbn-products.

| Method | Valid | Test | Memory | Time | Total |
|---|---|---|---|---|---|
| Supervised GCN | 92.0±0.0 | 75.6±0.2 | - | - | |
| MLP | 75.5±0.0 | 61.1±0.0 | - | - | |
| Node2vec | 70.0±0.0 | 68.8±0.0 | - | - | |
| BGRL (100 epos) | 78.1±2.1 | 64.0±1.6 | 29,303MB | 53m16s | 5,326m40s |
| GBT (100 epos) | 85.0±0.1 | 70.5±0.4 | 20,419MB | 48m38s | 4,863m20s |
| GGD(1 epo) | 90.9±0.5 | **75.7**±0.4 | 4,391MB\|78.5% | 12m46s | 12m46s\|381× |

of this hyperparameter specified in GBT [14]. For ogbn-products, we are 381 × faster than GBT [14] on total training time. Notably, our performance is significantly higher than GCL baselines using 100 epochs (i.e., 6% and 5.2% improvement on GBT [14] in validation and test set, respectively) with only one epoch training in this dataset. In addition, we compare the convergence speed among `GGD`, BGRL [15] and GBT [14] on ogbn-arxiv and ogbn-products, which are shown in Figure 4. For ogbn-arxiv, BGRL [15] is running using batched processing with neighbour sampling. This figure shows the preeminence of our `GGD` in convergence speed as `GGD` can be well-trained with only one epoch (i.e., reaching the peak model performance in the first epoch and staying stable with increased epochs). In contrast, the other two baselines require comparatively much more epochs to gradually improve their performance. Compared with GCL baselines, `GGD` achieves much faster convergence via Group Discrimination. We conjecture this is because GD-based method focuses on the general edge distribution of graphs instead of node-specific information. Inversely, GCL methods can suffer from convergence inefficiency as they may be easily distracted from too-detailed node-specific information during training.

**ogbn-papers100M.** We further compare `GGD` with BGRL [15] and GBT [14] on ogbn-papers100M, the largest OGB dataset with billion scale edges. Other self-supervised learning algorithms such as DGI [10] and GMI [12] fail to scale to such a large graph with a reasonable batch size (i.e., 256). We only report the performance of each algorithm after a single epoch of training in Table 10 due to the extreme scale of the dataset and the limitation of our available re-

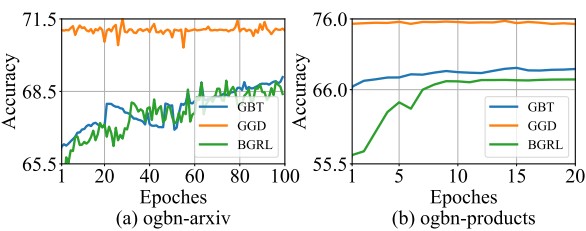

Figure 4: Convergence speed comparison among `GGD`, BGRL[15] and GBT [14]. X-axis means number of epochs, while Y-axis represents the accuracy on test set.

sources. From the table, we can observe that `GGD` outperforms the two GCL counterparts, BGRL [15] and GBT [14] in both accuracy and efficiency. Specifically, `GGD` achieves 60.2 in accuracy while BGRL and GBT reach 59.3 and 58.9 in test set, respectively. With only one epoch, these two algorithms may not be well trained. However, training each epoch of these two requires over 1 day and if we would like to train them for 100 epochs, then we will need 100+ GPU days, which is prohibitively impractical for general practitioners. In contrast, `GGD` can be trained in about 9 hours to achieve a good result for this dataset, which is more appealing in practice.

# 6 Explore Group Discrimination

In this section, we explore the corruption technique in `GGD` and provide the theoretical analysis of group discrimination.

## 6.1 Exploring Corruption

Firstly, we explore the corruption technique used in DGI [10] and MVGRL [11], which is shown in Figure 5. These two studies corrupt the topology of a given graph $\mathcal{G}$ by shuffling the feature matrix $\mathbf{X}$. This is because by changing the node order of $\mathbf{X}$, the neighbouring structure of $\mathcal{G}$ is completely changed, e.g., neighbours of node $a$ become node $b$ neighbours.

Table 10: Node classification result and efficiency comparison on ogbn-papers100M.

| Method | Validation | Test | Memory | Time |
|---|---|---|---|---|
| Supervised SGC | 63.3±0.2 | 66.5±0.2 | - | - |
| MLP | 47.2±0.3 | 49.6±0.3 | - | - |
| Node2vec | 55.6±0.0 | 58.1±0.0 | - | - |
| BGRL (1 epoch) | 59.3±0.5 | 62.1±0.3 | 14,057MB | 26h28m |
| GBT (1 epoch) | 58.9±0.4 | 61.5±0.5 | 13,185MB | 24h38m |
| GGD(1 epoch) | 60.2±0.3 | 63.5±0.5 | 4,105MB\|68.9% | 9h15m\|2.7× |

With the corruption technique, negative samples in the negative group are generated with incorrect edges. Thus, by discriminating the positive group (i.e., nodes generated with ground truth edges) and the negative group, we conjecture the model can distil valuable signals by learning how to identify nodes generated with correct topology and output effective node embeddings. To provide explanation to this, we present the theoretical analysis of group discrimination in the following section.

### 6.1.1 Theoretical Analysis of Group Discrimination

Group discrimination is learning to avoid making 'mistakes' (i.e., bias the encoder towards avoiding mistaken samples). To explain this point, we first present Theorem 1 and then provide an intuitive explanation for group discrimination.

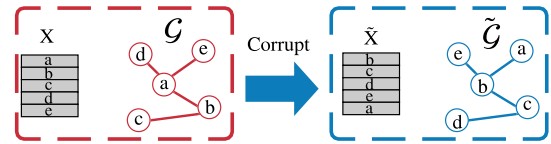

Figure 5: Corruption technique in DGI and MVGRL.

**Theorem 1** *Given a graph $\mathcal{G}$, a corrupted graph $\tilde{\mathcal{G}}$, and a encoding network $g(\cdot)$, we consider the distribution of positive embeddings $g(\mathcal{G})$ as $P_{pos}$ and negative embeddings $g(\tilde{\mathcal{G}})$ as $P_{neg}$. Optimising the group discrimination loss is equivalent to maximising the Jensen-Shannon divergence between $P_{pos}$ and $P_{neg}$.*

The proof for Theorem 1 is presented in Appendix A.2. From the theorem above, we can see maximising the group discrimination loss $\mathcal{L}$ is the same as maximising $JS(P_{pos} \parallel P_{neg})$, where $JS$ represents the Jenson-Shannon divergence. Thus, by optimising the loss $\mathcal{L}$, $P_{pos}$ and $P_{neg}$ tend to be separated. As a result, group discrimination is intuitively learning to avoid making 'mistakes' (i.e., bias the encoder towards avoiding mistaken samples) as shown in Figure 6. This is because by separating $P_{pos}$ and $P_{neg}$, $P_{pos}$ can gradually become similar to $P_{optimal}$, the optimal distribution for node embeddings. As $P_{optimal}$, is disjoint with $P_{neg}$, if the generated embeddings can avoid being similar to out-of-distribution samples, i.e., negative samples, it can be ideally closer to $P_{optimal}$. Therefore, the trained model can improve the quality of generated node embeddings for node samples.

## 7   Future Work

In this paper, we have introduced a new self-supervised GRL paradigm: Group Discrimination, which achieves the same level of performance as GCL methods with much less resource consumption (i.e., training time and memory). Some limitations of this work are we still have not explored some questions for GD. For example, can we extend the current binary Group Discrimination scheme (i.e., classifying nodes generated with different topology) to discrimination among multiple groups? Are there any other corruption tech-

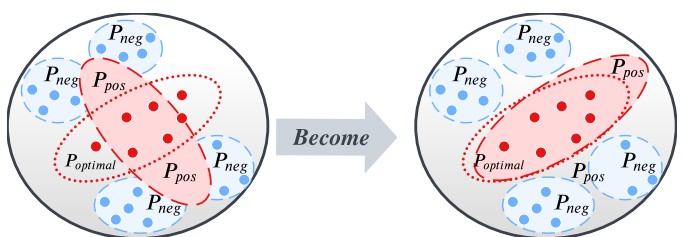

Figure 6: $P_{optimal}$ is the optimal distribution for node embeddings, $P_{pos}$ is the distribution of positive samples, $P_{neg}$ is the distribution of negative samples, blue nodes represent negative samples, and red nodes are samples in the optimal distribution. At the beginning, $P_{pos}$ is overlapped with $P_{neg}$. Then, $P_{pos}$ is gradually separated from $P_{neg}$ and ideally become closer to $P_{optimal}$.

nique to create a more difficult negative group for discrimination? More importantly, with the extremely efficient property, GD has the potential to be deployed to various real-world applications, e.g., recommendation systems, which have limited labelling information and desire fast computation with limited resources.

## Acknowledgments and Disclosure of Funding

This research was partially supported by an Australian Research Council (ARC) Future Fellowship (FT210100097).

This work is supported in part by NSF under grants III-1763325, III-1909323, III-2106758, and SaTC-1930941.

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
