## A  Appendix A

### A.1  Proof of Proposition 1

*Proof.* To prove Proposition 1, given a graph $\mathcal{G} = \{\mathbf{X}, \mathbf{A}\}$, where $\mathbf{X} \in \mathbb{R}^{N \times D}$, $\mathbf{A} \in \mathbb{R}^{N \times N}$ and a GNN encoder $g(\cdot)$. We consider $g(\cdot)$ as a one-layer GCN, which can be expressed as:

$$\mathbf{H} = \sigma(\hat{\mathbf{D}}^{-\frac{1}{2}}\hat{\mathbf{A}}\hat{\mathbf{D}}^{-\frac{1}{2}}\mathbf{Z}\mathbf{W}), \tag{5}$$

where $\sigma(\cdot)$ is non-linear activation function, $\mathbf{H} \in \mathbb{R}^{N \times D'}$ is the output embedding, $\mathbf{Z} = norm(\mathbf{X})$ ($norm(\cdot)$ is row normalisation), $\mathbf{W} \in \mathbb{R}^{D \times D'}$ is learnable weight matrix, $\hat{\mathbf{A}} = \mathbf{A} + \mathbf{I}$ ($\mathbf{I}$ is identity matrix), and $\hat{\mathbf{D}}$ is the degree matrix for $\hat{\mathbf{A}}$.

We consider $\tilde{\mathbf{A}} = \hat{\mathbf{D}}^{-\frac{1}{2}}\hat{\mathbf{A}}\hat{\mathbf{D}}^{-\frac{1}{2}}$ and $\mathbf{V} = \tilde{\mathbf{A}}\mathbf{Z}$, where elements $v_{ik} \in \mathbf{V}$, the $i$-th row and the $k$-th column element of $\mathbf{V}$, are definite. After that, we conduct $\mathbf{P} = \mathbf{VW}$, where $\mathbf{P} \in \mathbb{R}^{N \times D'}$ and $\mathbf{W}$ is Xavier initialised.

An elment $w_{mk} \sim U(-a, a)$ of $\mathbf{W}$ is the $m$-th row and the $k$-th column of $\mathbf{W}$, where $U(-a, a)$ is a uniform distribution. Here $a = \alpha \times \sqrt{\frac{6}{D+D'}}$, where $D'$ is the hidden dimension, and $\alpha$ is a hyperparameter defaultly set to 1. Then, we know the mean value of $w_{mk}$ distribution is 0 and its standard deviation is $\frac{2}{D+D'}$.

For any $p_{ik} \in \mathbf{P}$, which is the $i$-th row and $k$-th column of $\mathbf{P}$, it can be calculated as:

$$p_{ik} = \sum_{m}^{D} v_{im}w_{mk}, \tag{6}$$

where $w_{mk} \in \mathbf{W}$ is the $m$-th row and the $k$-th column of $\mathbf{W}$, and $v_{im} \in \mathbf{V}$ is the $i$-th row and $m$-th column of $\mathbf{V}$. According to the analysis on the weighted sum of uniform random variables [33], we have the mean $\mu$ and the standard deviation $\delta$ of $p_{ik}$:

$$\mu = 0, \delta = \sqrt{\frac{2}{D+D'}\sum_{m=1}^{D}v_{im}^2}. \tag{7}$$

Here we assume $p_{ik} \sim N(\mu, \delta^2)$ and use $[\mu - c\delta, \mu + c\delta]$ to approximate the value scope of $p_{ik}$, where $c$ is a parameter that controls the precision of approximation.

**Case 1 (ReLU/Leaky ReLu/PReLu).** In this case, we consider ReLu $\sigma_{ReLu}(\cdot)$, Leaky ReLu $\sigma_{LReLu}(\cdot)$, and PReLu $\sigma_{PReLu}(\cdot)$ as non-linear activation for the GNN encoder $g(\cdot)$. Here we take $h_{ik} = \sigma_{ReLu}(p_{ik})$ ($h_{ik} \in \mathbf{H}$) as an example to show that the value range of $\sigma_{sig}(\sigma_{ReLu}(p_{ik}))$ is bounded with $\delta$. Please noted that, in practice, $\mathbf{H}$ will be averaged to a summary vector $\mathbf{s}$ and input to $\sigma_{sig}(\cdot)$. As $\sigma_{sig}(s)$ ($s \in \mathbf{s}$) shares the same value range as $\sigma_{sig}(\sigma_{ReLu}(p_{ik}))$, for simplicity, we conducted analysis with $\sigma_{ReLu}(p_{ik})$ instead.

Given $p_{ik}$ as input, the value range of $\sigma_{ReLu}(p_{ik}) = max(0, p_{ik})$ is $[0, c\delta]$. Similarly, we can also obtain $\sigma_{LReLu}(x) = max(0.01 * x, x) \in [-0.01c\delta, c\delta]$ and $\sigma_{PReLu}(x) = max(0, x) + b * min(0, x) \in [-bc\delta, c\delta]$(when $b \geq 0$) or $[0, max(-bc\delta, c\delta)]$(when $b < 0$). Given $\sigma_{ReLu}(p_{ik}) \in [0, c\delta]$, the infimum of $\sigma_{sig}(\sigma_{ReLu}(p_{ik}))$ is $\frac{1}{2}$ since sigmoid activation is a monotone increasing function. To estimate the value range of final outputs, here we use the Taylor series of $\sigma_{sig}(\cdot)$ at 0 to estimate the upper bound [34], i.e.,

$$\sigma_{sig}(x) \approx \sigma_{sig}(0) + \sigma'_{sig}(0)x + \frac{1}{2}\sigma''_{sig}(0)x^2 + \cdots. \tag{8}$$

Since $\sigma_{sig}(0) = \frac{1}{2}$, $\sigma'_{sig}(x) = \sigma_{sig}(x)(1 - \sigma_{sig}(x))$, $\sigma''_{sig}(x) = \sigma_{sig}(x)(1 - \sigma_{sig}(x))(1 - 2\sigma_{sig}(x))$, and $\frac{\partial^n}{\partial x^n}\sigma_{sig}(x) = \sigma_{sig}(x)\prod_{i=1}^{n}(1 - i\sigma_{sig}(x))$, Equation (8) can be expressed as:

$$\sigma_{sig}(x) \approx \sigma_{sig}(0) + \sigma'_{sig}(0)x = \frac{1}{2} + \frac{1}{4}x. \tag{9}$$

Thus, the value range of $\sigma_{sig}(\sigma_{ReLu}(p_{ik}))$ is $[\frac{1}{2}, \frac{1}{2} + \frac{c\delta}{4}]$, which indicates $\sigma_{sig}(\sigma_{ReLu}(p_{ik})) \to \frac{1}{2}$ when $\delta \to 0$.

**Case 2 (Sigmoid).** In this case, we consider Sigmoid $\sigma_{sig}(\cdot)$ as non-linear activation in $g(\cdot)$. We can also obtain $\sigma_{sig}(p_{ik}) \in [\frac{1}{2} - \frac{c\delta}{4}, \frac{1}{2} + \frac{c\delta}{4}]$ through Equation (9) and $\sigma_{sig}(p_{ik}) \to \frac{1}{2}$ when $\delta \to 0$, which also indicates $\sigma_{sig}(\sigma_{sig}(p_{ik})) \to 0.62$ (i.e., $\sigma_{sig}(\frac{1}{2})$) when $\delta \to 0$. In our experiments, we observe that $\delta$ owns a almost zero positive value, which results in our observations in Table 2.

## A.2 Proof for Theorem 1

*Proof.* The following proof is inspired by the theoretical proof in GAN [35]. During training, we can rewrite the group discrimination loss in Equation 3 into the following objective (maximise to optimise):

$$\mathcal{L} = \mathbb{E}_{\mathbf{h}\sim P_{pos}}log(agg(\mathbf{h})) + \mathbb{E}_{\mathbf{h}\sim P_{neg}}log(1 - agg(\mathbf{h})),$$
$$= \int_{\mathbf{h}} P_{pos}(\mathbf{h})log(agg(\mathbf{h}))d\mathbf{h} + \int_{\mathbf{h}} P_{neg}(\mathbf{h})log(1 - agg(\mathbf{h}))d\mathbf{h}, \tag{10}$$

where $agg(\cdot)$ is the aggregation function to turn $\mathbf{h}$ into the $1 \times 1$ prediction, $P_{pos}$ are the distribution of positive embeddings, $P_{neg}$ are the distribution of negative embeddings. As our objective here is to maximise $\mathcal{L}$, and $P_{pos}(\mathbf{h}) > 0; P_{neg}(\mathbf{h}) > 0$, we can obtain the optimal solution for $agg(\mathbf{h})$ is $\frac{P_{pos}(\mathbf{h})}{P_{pos}(\mathbf{h})+P_{neg}(\mathbf{h})}$. This is because for any $(a,b) \in \mathbb{R}^2 \backslash \{0,0\}$, the maximum of a function $y = alog(x)+blog(1-x)$ is achieved at $\frac{a}{a+b}$ [35]. By replacing $agg(\mathbf{h})$ with $\frac{P_{pos}(\mathbf{h})}{P_{pos}(\mathbf{h})+P_{neg}(\mathbf{h})}$ in Equation 10, we can obtain:

$$
\begin{aligned}
\mathcal{L} &= \mathbb{E}_{\mathbf{h} \sim P_{pos}} log(\frac{P_{pos}(\mathbf{h})}{P_{pos}(\mathbf{h}) + P_{neg}(\mathbf{h})}) + \mathbb{E}_{\mathbf{h} \sim P_{neg}} log(1 - \frac{P_{pos}(\mathbf{h})}{P_{pos}(\mathbf{h}) + P_{neg}(\mathbf{h})}), \\
&= \mathbb{E}_{\mathbf{h} \sim P_{pos}} log(\frac{P_{pos}(\mathbf{h})}{P_{pos}(\mathbf{h}) + P_{neg}(\mathbf{h})}) + \mathbb{E}_{\mathbf{h} \sim P_{neg}} log(\frac{P_{neg}(\mathbf{h})}{P_{pos}(\mathbf{h}) + P_{neg}(\mathbf{h})}).
\end{aligned}
\tag{11}
$$

From the equation above, we can see it looks similar to the Jensen-Shannon divergence between two distribution $P_1$ and $P_2$:

$$
JS(P_1 \parallel P_2) = \frac{1}{2}\mathbb{E}_{\mathbf{h} \sim P_1} log(\frac{\frac{P_1}{P_1+P_2}}{2}) + \frac{1}{2}\mathbb{E}_{\mathbf{h} \sim P_2} log(\frac{\frac{P_2}{P_1+P_2}}{2}).
\tag{12}
$$

Thus, we can rewrite Equation 11 as:

$$
\begin{aligned}
\mathcal{L} &= \mathbb{E}_{\mathbf{h} \sim P_{pos}} log(\frac{\frac{P_{pos}(\mathbf{h})}{P_{pos}(\mathbf{h})+P_{neg}(\mathbf{h})}}{2}) + \mathbb{E}_{\mathbf{h} \sim P_{neg}} log(\frac{\frac{P_{neg}(\mathbf{h})}{P_{pos}(\mathbf{h})+P_{neg}(\mathbf{h})}}{2}) - 2log2, \\
&= 2JS(P_{pos} \parallel P_{neg}) - 2log2,
\end{aligned}
\tag{13}
$$

where we can see maximising $\mathcal{L}$ is the same as maximising $JS(P_{pos} \parallel P_{neg})$. Thus, by optimising $\mathcal{L}$, $P_{pos}$ and $P_{neg}$ tend to be separated.

### A.2.1 Connection with DGI

In this section, by building a connection with DGI, we explain our theoretical motivation for simplifying DGI objective to group discrimination loss and why group discrimination is efficient in computation time. We first present the Lemma 1, which is used in DGI [10]:

**Lemma 1** *Define $\{\mathbf{H}^g\}_{g=1}^{|\mathbf{H}|}$ as a set of node embeddings drawn from distribution of graphs, $p(\mathbf{H})$, where $|\mathbf{H}|$ is finite number of elements, and $p(\mathbf{H}^g) = p(\mathbf{H}^{g'}), \forall g, g'$. $\mathcal{R}(\cdot)$ is a deterministic readout function, which takes $\mathbf{H}^g$ as input and output the summary vector of the g-th graph, $\mathbf{s}^g$. $\mathbf{s}^g$ follows a marginal distribution $p(\mathbf{s})$. Then, we assume $\mathcal{R}(\cdot)$ is injective and class balance, the upper bound of the error rate for the optimal classifier between the joint distribution $p(\mathbf{H}, \mathbf{s})$ and $p(\mathbf{H})p(\mathbf{s})$ is capped at $Er^* = \frac{1}{2}\sum_{g=1}^{|\mathbf{H}|} p(\mathbf{s}^g)^2$.*

Based on our analysis in Section 2.1, we assume $\mathbf{s}$ is a constant summary vector $\epsilon \boldsymbol{I}$, where $\epsilon$ is the constant in $\mathbf{s}$. In addition, we assume $\epsilon$ in $\mathbf{s}$ is independent from $p(\mathbf{H})$. Then, we can derive the following lemma:

**Lemma 2** *We assume $\mathbf{s}$ is a constant summary vector $\epsilon \boldsymbol{I}$ and $\epsilon$ of $\mathbf{s}$ is independent from $p(\mathbf{H})$, the error rate for the optimal classifier between the joint distribution $p(\mathbf{H}, \mathbf{s})$ and the product of marginals $p(\mathbf{H})p(\mathbf{s})$ is $Er^* = \frac{1}{2}$.*

*Proof.* As $\epsilon$ is independent from $p(\mathbf{H})$, we can see $p(\mathbf{s})$ is independent from $p(\mathbf{H})$. Thus, the joint distribution $p(\mathbf{H}, \mathbf{s})$ equals to the product of marginals $p(\mathbf{H})p(\mathbf{s})$. As a result, every sample from the joint is also a sample from the product of marginals. In this case, no classifier performs better than random guess, i.e., no classifier can discriminate samples from $p(\mathbf{H}, \mathbf{s})$ and $p(\mathbf{H})p(\mathbf{s})$ in this case. Therefore, we prove that $Er^* = \frac{1}{2}$ [10].

Then we present Theorem 2, which is presented in DGI[10]:

**Theorem 2** *Define $\mathbf{s}^*$ as the optimal summary vector under the classification error of an optimal classifier between $p(\mathbf{H}, \mathbf{s})$ and $p(\mathbf{H})p(\mathbf{s})$. $\mathbf{s}^* = argmax_{\mathbf{s}} MI(\mathbf{H}; \mathbf{s})$, where MI stands for mutual information.*

Based on Theorem 2, in DGI, they claim that for finite input sets and appropriate deterministic functions, minimising the classification error in the discriminator $\mathcal{D}(\cdot)$ (as shown in Equation 2) can be used to maximise the MI between the input and output of $\mathcal{R}(\cdot)$. However, under the aforementioned assumptions, the error rate $Er^*$ is a constant, and it is not practical to minimise the classification error. In addition, as $p(\mathbf{s})$ is independent from $p(\mathbf{H})$, we know $MI(\mathbf{H}; \mathbf{s}) = 0$. Thus, we can see these findings contradict to Theorem 2.

Instead of maximising the $MI(\mathbf{H}; \mathbf{s})$, in this case, the discriminator $\mathcal{D}(\cdot)$ is responsible for maximising the similarity between positive embeddings and the constant summary vector $\mathbf{s}$, while minimising the similarity between negative embeddings and $\mathbf{s}$. This operation is equivalent to maximising the Jensen-Shannon divergence between the distribution of positive embeddings and negative embeddings. We show a theorem to explain this as follows:

**Theorem 3** *Assuming* $\mathbf{s}$ *is a constant summary vector* $\epsilon \mathbf{I}$ *and* $\epsilon$ *of* $\mathbf{s}$ *is independent from* $p(\mathbf{H})$*. Given a graph* $\mathcal{G}$*, a corrupted graph* $\tilde{\mathcal{G}}$*, and a GNN encoder* $g_\theta(\cdot)$*, we consider the distribution of positive embeddings* $g_\theta(\mathcal{G})$ *as* $P_{pos}^{\mathbf{h}}$ *and negative embeddings* $g_\theta(\tilde{\mathcal{G}})$ *as* $P_{neg}^{\mathbf{h}}$*. Optimising the DGI loss is equivalent to maximising the Jensen-Shannon divergence between* $P_{pos}^{\hat{\mathbf{h}}}$ *and* $P_{neg}^{\hat{\mathbf{h}}}$*, where* $\hat{\mathbf{h}}$ *is linearly transformed* $\mathbf{h}$*.*

*Proof.* We first present the DGI loss with a constant summary vector:

$$
\begin{aligned}
\mathcal{L} &= \mathbb{E}_{\mathbf{h} \sim P_{pos}^{\mathbf{h}}} log \mathcal{D}(\mathbf{h}, \mathbf{s}) + \mathbb{E}_{\mathbf{h} \sim P_{neg}^{\mathbf{h}}} log(1 - \mathcal{D}(\mathbf{h}, \mathbf{s})), \\
&= \mathbb{E}_{\mathbf{h} \sim P_{pos}^{\mathbf{h}}} log(\mathbf{h} \cdot \mathbf{W} \cdot \mathbf{s}) + \mathbb{E}_{\mathbf{h} \sim P_{neg}^{\mathbf{h}}} log(1 - \mathbf{h} \cdot \mathbf{W} \cdot \mathbf{s}), \\
&= \mathbb{E}_{\mathbf{h} \sim P_{pos}^{\mathbf{h}}} log(\mathbf{h} \cdot \mathbf{W} \cdot \epsilon) + \mathbb{E}_{\mathbf{h} \sim P_{neg}^{\mathbf{h}}} log(1 - \mathbf{h} \cdot \mathbf{W} \cdot \epsilon),
\end{aligned}
\tag{14}
$$

where $\mathbf{h}$ is node embedding and $\mathbf{W}$ is learnable weight matrix. Here, we consider $\mathbf{h} \cdot \mathbf{W}$ (i.e., linearly transformed $\mathbf{h}$) as $\hat{\mathbf{h}}$. Then, we consider the distribution of $\hat{\mathbf{h}}$ generated with positive samples $\mathbf{h}$ as $P^{\hat{\mathbf{h}}_{pos}}$ and the distribution of $\hat{\mathbf{h}}$ with negative samples $\mathbf{h}$ as $P^{\hat{\mathbf{h}}_{pos}}$. Then, the equation becomes:

$$
\begin{aligned}
\mathcal{L} &= \mathbb{E}_{\hat{\mathbf{h}} \sim P_{pos}^{\hat{\mathbf{h}}}} log(sum(\epsilon \hat{\mathbf{h}})) + \mathbb{E}_{\hat{\mathbf{h}} \sim P_{neg}^{\hat{\mathbf{h}}}} log(1 - sum(\epsilon \hat{\mathbf{h}})), \\
&= \mathbb{E}_{\hat{\mathbf{h}} \sim P_{pos}^{\hat{\mathbf{h}}}} log(\epsilon \cdot agg(\hat{\mathbf{h}})) + \mathbb{E}_{\hat{\mathbf{h}} \sim P_{neg}^{\hat{\mathbf{h}}}} log(1 - \epsilon \cdot agg(\hat{\mathbf{h}})),
\end{aligned}
\tag{15}
$$

the above equation is very similar to Equation 11 and the only difference is there is a $\epsilon$ multiply with the $agg(\cdot)$ output. Here, $agg(\cdot)$ is summation. Thus, the proof for Theorem 2 still holds and prove Theorem 3.

Based on Theorem 3, we can see group discrimination without the summary vector is doing the same thing as DGI with a constant summary vector (i.e., separating positive and negative distribution). Thus, we are motivated to remove the summary vector from the loss and proposed the group discrimination loss in Equation 3.

Instead of relying on a summary vector $\mathbf{s}$ to discriminate positive and negative samples in $\mathbf{H}$ (i.e., calculating their similarity with the summary vector), we directly use a binary cross entropy loss to classify these samples. Removing the summary vector $\mathbf{s}$ is beneficial to the computation efficiency because it eases the burden of gradient computation, e.g., to compute the gradient for $\mathbf{s}$, we need to store and use all the parameters in the model to conduct backward propagation. However, in group discrimination, we do not need the summary vector and only aggregate node embeddings to obtain prediction.

### A.3 Evaluation on aggregation function

To explore the effect of other aggregation functions, we replace the summation function in Equation 1 with other aggregation methods, including mean-, minimum-, maximum- pooling and linear aggregation. We report the experiment results (i.e., averaged accuracy on five runs) in Table 11. The table shows that replacing the summation with other aggregation methods still works, while summation and linear aggregation achieve comparatively better performance.

Table 11: The experiment result on three datasets with different aggregation function on node embeddings.

| Method | Cora | CiteSeer | PubMed |
|---|---|---|---|
| Sum | 82.5 ±0.2 | 71.7 ±0.6 | 77.7 ±0.5 |
| Mean | 81.8 ±0.5 | 71.8 ±1.1 | 76.5 ±1.2 |
| Min | 80.4 ±1.3 | 61.7 ±1.8 | 70.1 ±1.9 |
| Max | 71.4 ±1.2 | 65.3 ±1.4 | 70.2 ±2.8 |
| linear | 82.2 ±0.4 | 72.1 ±0.7 | 77.9 ±0.5 |

### A.4 Rethinking MVGRL

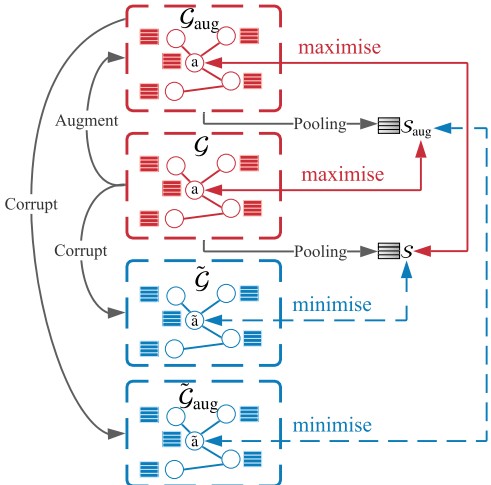

Figure 7: The architecture of MVGRL. Here augment means augmentation. $\mathbf{s}$ is the summary vector based on $\mathcal{G}$, and $\mathbf{s}_{aug}$ is the summary vector based on the augmented graph $\mathcal{G}_{aug}$.

Extending the architecture of DGI, MVGRL resorts to multi-view contrastiveness via additional augmentation. Specifically, as shown in Figure 7, it first uses the diffusion augmentation to create $\mathcal{G}_{aug}$. Then, it corrupts $\mathcal{G}$ and $\mathcal{G}_{aug}$ to generate negative samples $\tilde{\mathcal{G}}$ and $\tilde{\mathcal{G}}_{aug}$. To build contrastiveness, MVGRL also generates two summary vectors $\mathbf{s}_{aug}$ and $\mathbf{s}$ by averaging all embeddings in $\mathcal{G}_{aug}$ and $\mathcal{G}$, respectively. Based on the design of MVGRL, the model training is driven by mutual information maximisation between an anchor node embedding and its corresponding augmented summary vector. However, MVGRL has the same technical error in their official JSD-based implementation as DGI, which makes it also becomes a group-discrimination-based approach.

Similar to Equation 3 of DGI, the proposed loss in MVGRL can also be rewritten as a binary cross entropy loss:

$$\mathcal{L}_{MVGRL} = -\frac{1}{4N}(\sum_{i=1}^{4N} y_i \log \hat{y}_i + (1 - y_i)\log(1 - \hat{y}_i)), \tag{16}$$

here the number of nodes is increased to $4N$ as we include both nodes in $\mathcal{G}_{aug}$ and $\tilde{\mathcal{G}}_{aug}$ as data samples. The indicator $y_i$ for $\mathcal{G}$ and $\mathcal{G}_{aug}$ are 1, while $\tilde{\mathcal{G}}$ and $\tilde{\mathcal{G}}_{aug}$ are considered as negative samples (i.e., the indicator $y_i$ for them are 0). To explore why MVGRL can achieve a better performance than DGI, we replace the original MVGRL loss with Equation 16 and conduct ablation study of $MVGRL_{bce}$ by removing different set of data samples in Equation 16, and report the experiment result in Table 12. From the table, the performance of $MVGRL_{bce}$ is on par with $MVGRL$, which reconfirms the effectiveness of using the BCE loss. Also, we can observe that including $\mathcal{G}_{aug}$ and $\tilde{\mathcal{G}}_{aug}$ is the key of MVGRL surpassing DGI. With $\mathcal{G}_{aug}$ and $\tilde{\mathcal{G}}_{aug}$, the model performance of $MVGRL_{bce} w/o \mathcal{G}_{aug}$ and $\tilde{\mathcal{G}}_{aug}$ is improved from 82.2 to 83.1. We conjecture this is because, with the diffusion augmentation, MVGRL is trained with the additional global information provided by the diffused view $\mathcal{G}_{aug}$. However, the diffusion augmentation involves expensive matrix inversion computation and significantly densifies the given graph, which requires much more memory and time to store and process than the original view. This can hinder the model from extending to large-scale datasets [21].

### A.5 Complexity Analysis

The time complexity of our method consists of two components: the *siamese GNN* and the *loss computation*. Existing self-supervised baselines share similar time complexity for the first component. In GGD, given a graph $\mathcal{G} = \{\mathbf{X} \in \mathbb{R}^{N \times D}, \mathbf{A} \in \mathbb{R}^{N \times N}\}$ in the sparse format, taking a $L$-layer

Table 12: The ablation study of MVGRL from the perspective of Group Discrimination.

| Method | Cora | CiteSeer | PubMed |
|---|---|---|---|
| $MVGRL$ | 82.9 ±0.9 | 72.6 ±0.8 | 78.8 ±0.6 |
| $MVGRL_{BCE}$ | 83.1 ±0.6 | 72.8 ±0.5 | 79.1 ±1.1 |
| $MVGRL_{BCE}$ w/o $\mathcal{G}_{aug}$ | 81.2 ±0.8 | 52.8 ±3.1 | 76.6 ±1.3 |
| $MVGRL_{BCE}$ w/o $\mathcal{G}$ | 82.1 ±0.6 | 71.8 ±1.1 | 77.1 ±1.2 |
| $MVGRL_{BCE}$ w/o $\tilde{\mathcal{G}}_{aug}$ | 81.1 ±0.8 | 56.7 ±2.1 | 74.9 ±1.3 |
| $MVGRL_{BCE}$ w/o $\tilde{\mathcal{G}}$ | 82.7 ±0.9 | 72.0 ±0.9 | 78.6 ±0.9 |
| $MVGRL_{BCE}$ w/o $\mathcal{G}_{aug}$ and $\tilde{\mathcal{G}}_{aug}$ | 82.2 ±0.6 | 71.8 ±1.0 | 77.0 ±0.8 |
| $MVGRL_{BCE}$ w/o $\mathcal{G}$ and $\tilde{\mathcal{G}}$ | 83.1 ±0.6 | 72.6 ±0.6 | 78.5 ±1.4 |

GCN [19] encoder as an example, the time complexity is $O(LND + LND^2)$ [3]. As we need to process both the augmented graph $\hat{\mathcal{G}}$ and the corrupted graph $\tilde{\mathcal{G}}$, GGD requires the encoder computation twice. Then, the projector network (i.e., MLP) with $K$ linear layers will be applied to the encoder output, which takes $O(ND^2)$ for each layer in computation [4]. Before group discrimination, we aggregate the generated embedding with aggregation techniques. Here we take the simple summation, which consumes $O(ND)$ as example. For the loss computation, we use the BCE loss, i.e., Equation 3, to category summarised node embeddings, i.e., scalars. The time complexity of this final step is $O(2N)$ (i.e., processing all data samples from the positive and negative groups). Ignoring the computation cost of the augmentation, the overall time complexity of GGD for computing a graph $\mathcal{G}$ is $O(2(LND + LND^2 + KND^2 + ND + N)) \rightarrow O(ND(L + LD + KD))$, where we can see the time complexity is mainly contributed by the siamese GNN. Also, our model scales linearly w.r.t. the number of nodes $N$.

## A.6 The power of Graphs

To show the easiness of computation for the power of graphs, we conduct an experiment to evaluate the time consumption for graph power computation on eight datasets, whose statistics are shown in Appendix A.7. Specifically, we set the hidden size of $\mathbf{H}_\theta$ to 256, and $n$ is fixed to 10 for all datasets. The experiment results are shown as below:

Table 13: Graph power computation time in seconds on eight benchmark datasets. The experiment is conducted using CPU: Intel Xeon Gold 5320. 'Cite' , 'Comp', 'Photo', 'Arxiv', 'Products', 'Papers' means Citeseer, Amazon Computer, Amazon Photo, ogbn-arxiv, ogbn-products and ogbn-papers100M.

| Cora | Cite | PubMed | Comp | Photo | Arxiv | Products | Papers |
|---|---|---|---|---|---|---|---|
| 5.4e-3 | 7.3e-3 | 9.8e-3 | 1.2e-2 | 8.5e-3 | 2.2e-2 | 24.5 | 208.8 |

This table shows that the computation of graph power is very trivial on small and medium size graphs, e.g., ogbn-arxiv, which has million of edges, consuming only 0.22 seconds. Extending to an extremely large graph, ogbn-papers100M, which has over 1 billion edges and 11 million nodes, the computation only requires 209 seconds (i.e., around three minutes), which is acceptable considering the sheer size of the dataset.

## A.7 Dataset Statistics

The following table presents the statistics of eight benchmark datasets including five small to medium -scale datasets and three large-scale datasets from OGB Graph Benchmark[29].

---

[3] Here we assume for simplicity that the graph is sparse with the number of edges $|E| = O(N)$.

[4] For simplicity we assume the hidden dimension size is $D$. In practice, the hidden dimension size $D'$ will be smaller than $D$.

Table 14: The statistics of eight benchmark datasets.

| Dataset | Nodes | Edges | Features | Classes |
|---|---|---|---|---|
| Cora | 2,708 | 5,429 | 1,433 | 7 |
| CiteSeer | 3,327 | 4,732 | 3,703 | 6 |
| PubMed | 19,717 | 44,338 | 500 | 3 |
| Amazon Computers | 13,752 | 245,861 | 767 | 10 |
| Amazon Photo | 7,650 | 119,081 | 745 | 8 |
| ogbn-arxiv | 169,343 | 1,166,243 | 128 | 40 |
| ogbn-products | 2,449,029 | 61,859,140 | 100 | 47 |
| ogbn-papers-100M | 111,059,956 | 1,615,685,872 | 100 | 172 |

## A.8 Experiment Settings & Computing Infrastructure

**Extending to Extremely Large Datasets.** Extending to extremely large graphs (i.e., ogbn-products and ogbn-papers100M), we adopt a simple neighbourhood sampling strategy introduced in Graph-Sage [23] to decouple model training from the sheer size of graphs. Specifically, we create a fixed size subgraph for each node, which is created by sampling a predefined number of neighbours in each convolution layer for sampled nodes. The same approach is employed in the testing phase to obtain final embeddings.

**General Parameter Settings.** In our experiment, we mainly tune four parameters for GGD ,which are learning rate, hidden size, number of convolution layers in the GNN encoder, and number of linear layers in the projector. For simplicity, we set the the power of a graph for global embedding generation fixed to 5 for all datasets (i.e., Equation 4). The parameter setting for each dataset is shown below:

Table 15: Parameter settings on eight datasets. 'lr' means the learning rate for pretraining, 'num-conv' and 'num-proj' represent number of convolution layers in GNNs and number of linear layers in projector, respectively.

| Dataset | lr | hidden | num-conv | num-proj |
|---|---|---|---|---|
| Cora | 1e-3 | 512 | 1 | 1 |
| CiteSeer | 1e-5 | 1024 | 1 | 1 |
| PubMed | 1e-3 | 1024 | 1 | 1 |
| Amazon Computers | 1e-3 | 1024 | 1 | 1 |
| Amazon Photo | 1e-3 | 512 | 1 | 1 |
| ogbn-arxiv | 5e-5 | 1500 | 3 | 1 |
| ogbn-products | 1e-4 | 1024 | 4 | 4 |
| ogbn-papers-100M | 1e-3 | 256 | 3 | 1 |

**Large-scale Datasets Parameter Settings.** To decouple model training from the scale of graphs, we adopt the neighbouring sampling technique, which has three parameters: batch size, sample size, and number of hops to be sampled. Batch size refers to the number of nodes to be processed in one parameter optimisation step. Sample size means the number of nodes to be sampled in each convolution layer, and the number of hops determines the scope of the neighbourhood for sampling. In GGD implementation, the batch size, sample size, and the number of hops are fixed to 2048, 12 and 3, respectively.

**Memory and Training Time Comparison.** As memory and training time are very sensitive to hyperparameters related to the structure of GNNs, including hidden size, number of convolution layers, and batch processing for large-scale datasets, e.g., batch size and number of neighbours sampled in each layer. Thus, in memory and training comparison, to be fair, we set all these parameters to be the same for all baselines and GGD. The specific parameter setting for each dataset is shown below:

Table 16: Parameter settings on eight datasets for memory and training time comparison.

| Dataset | hidden | num-conv | batch | num-neigh |
|---|---|---|---|---|
| Cora | 512 | 1 | - | - |
| CiteSeer | 512 | 1 | - | - |
| PubMed | 256 | 1 | - | - |
| Amazon Computers | 256 | 1 | - | - |
| Amazon Photo | 256 | 1 | - | - |
| ogbn-arxiv | 256 | 3 | - | - |
| ogbn-products | 256 | 3 | 512 | 10 |
| ogbn-papers-100M | 128 | 3 | 512 | 10 |

**Computing Infrastructure.** For experiments in Section 2, 3 and 6.1, they are conducted using Nvidia GRID T4 (16GB memory) and Intel Xeon Platinum 8260 with 8 core. For experiments on large-scale datasets (i.e.,ogbn-arxiv, ogbn-products and ogbn-papers100M), we use NVIDIA A40 (48GB memory) and Intel Xeon Gold 5320 with 13 cores.

### A.9 Algorithm

We have summarised the overall procedure of GGD in Algorithm 1 in **as follows**.

---

**Algorithm 1:** The Overall Procedure of GGD

**Input** : Input Graph $\mathcal{G} = \{\mathbf{X} \in \mathbb{R}^{N \times D}, \mathbf{A} \in \mathbb{R}^{N \times N}\}$; GNN encoder $g_\theta(\cdot)$; Projector $f_\theta(\cdot)$; Number of nodes $N$; Number of feature dimensions $D$; Number of hidden dimensions $D'$; Number of training epochs $T$

**Output** : Final representation $\mathbf{H}$

1    *//Model Training*;
2    **for** $t = 1$ *to* $T$ **do**
3       *//Augmentation*(optional);
4       Conduct feature dropout and edge dropout on $\mathcal{G}$ to obtain $\hat{\mathcal{G}} = \{\hat{\mathbf{X}}, \hat{\mathbf{A}}\}$;
5       *//Corruption*;
6       Corrupt $\hat{\mathcal{G}}$ to obtain $\tilde{\mathcal{G}} = \{\tilde{\mathbf{X}}, \tilde{\mathbf{A}}\}$;
7       *//Compute Encoding*;
8       Input $\hat{\mathcal{G}}$ and $\tilde{\mathcal{G}}$ to $g_\theta(\cdot)$ and $f_\theta(\cdot)$ to obtain graph embedding $\hat{\mathbf{H}}_\theta = f_\theta(g_\theta(\hat{\mathcal{G}}))$ and $\tilde{\mathbf{H}}_\theta = f_\theta(g_\theta(\tilde{\mathcal{G}}))$, respectively;
9       *//Aggregation*;
10      Concatenate $\hat{\mathbf{H}}_\theta$ and $\tilde{\mathbf{H}}_\theta$ to obtain $\bar{\mathbf{H}}_\theta$;
11      Conduct aggregation on $\bar{\mathbf{H}}_\theta \in \mathbb{R}^{2N \times D'}$ to obtain the prediction vector $\hat{\mathbf{y}}_\theta \in \mathbb{R}^{2N}$;
12      *//Compute Loss*;
13      Calculate loss $\mathcal{L} = -\frac{1}{2N}(\sum_{i=1}^{2N} y_i \log \hat{y}_i + (1 - y_i) \log(1 - \hat{y}_i))$, where $\hat{y}_i \in \mathbb{R}^1$ is the prediction for one node sample and $\hat{y}_i \in \hat{\mathbf{y}}_\theta$;
14      *//Update parameters*;
15      Update trainable parameters in $g_\theta(\cdot)$ and $f_\theta(\cdot)$;
16    *//Inference*;
17    Obtain local embedding $\mathbf{H}_\theta = f_\theta(g_\theta(\mathcal{G}))$ ;
18    Obtain global embedding $\mathbf{H}_\theta^{global} = \mathbf{A}^n \mathbf{H}_\theta$;
19    Obtain final embeddings for downstream tasks $\mathbf{H} = \mathbf{H}_\theta^{global} + \mathbf{H}_\theta$;

---

Specifically, we first conduct augmentation on $\mathcal{G}$ to obtain $\hat{\mathcal{G}}$. Then, $\hat{\mathcal{G}}$ is corrupted to generate the corrupted graph $\tilde{\mathcal{G}}$ . In the encoding phase, we feed $\hat{\mathcal{G}}$ and $\tilde{\mathcal{G}}$ to GNN encoder $g_\theta$ and projector $f_\theta$ to generate embeddings $\hat{\mathbf{H}}_\theta$ for positive samples and $\tilde{\mathbf{H}}_\theta$ for negative samples. After that, we obtain the binary classification result by aggregating the concatenation of $\hat{\mathbf{H}}_\theta$ and $\tilde{\mathbf{H}}_\theta$. The result is a $2N$ dimension vector, which can be used for calculating the loss with a BCE loss. Finally, based on the

calculated loss, trainable parameters in $g_\theta(\cdot)$ and $f_\theta(\cdot)$ can be updated. This training process will continue iteratively until we reach the predefined number of epochs $T$. When the training process is completed, we freeze the $g_\theta(\cdot)$ and $f_\theta(\cdot)$ and feed $\mathcal{G}$ to these two encoding components to obtain the local embedding, $\mathbf{H}_\theta$. Then, we obtain global embedding $\mathbf{H}_\theta^{global}$ with a global information injection operation. By summing $\mathbf{H}_\theta$ and $\mathbf{H}_\theta^{global}$, we can get the final embeddings $\mathbf{H}$.

## A.10 Ablation Study

In this section, we conduct an ablation study to evaluate the effectiveness of different components in GGD. Specifically, we evaluate three variants of GGD, including $\text{GGD}_{w/o\ aug}$, $\text{GGD}_{w/o\ proj}$, $\text{GGD}_{w/o\ power}$, which represent GGD without augmentation, the projector and the global information injection process respectively. The experiment results on five small to medium size datasets are presented in Table 17. From the table, we can see without any mentioned component, the performance of GGD degrades, which validates the effectiveness of these components. It is worth noting that even without the global information injection in the inference phase, $\text{GGD}_{w/o\ power}$ still achieves the highest performance in 4 out of 5 datasets compared with six self-supervised baselines. This indicates that even without global information injection, GGD is still effective.

Table 17: Ablation Study for GGD.

| Method | Cora | CiteSeer | PubMed | Comp | Photo |
|---|---|---|---|---|---|
| $\text{GGD}_{w/o\ aug}$ | 83.6±0.3 | 72.4±0.4 | 81.2±0.2 | 89.6±0.4 | 92.2 ±0.5 |
| $\text{GGD}_{w/o\ proj}$ | 83.0±0.5 | 72.5±0.4 | 81.1±0.4 | 89.4±0.5 | 91.6 ±0.5 |
| $\text{GGD}_{w/o\ power}$ | 83.0±0.5 | 72.5±0.4 | 80.1±0.4 | 89.9±0.6 | 91.6±0.4 |
| GGD | **83.9**±0.4 | **73.0**±0.6 | **81.3**±0.8 | **90.1**±0.9 | **92.5**±0.6 |