# OpenReview forum: "Rethinking and Scaling Up Graph Contrastive Learning: An Extremely Efficient Approach with Group Discrimination"
_NeurIPS.cc/2022/Conference — NeurIPS 2022 Accept_

### Official Review · Reviewer_5yox · 2022-07-08

**Rating:** 5
**Confidence:** 5
**Soundness:** 2 fair
**Presentation:** 3 good
**Contribution:** 2 fair

**Summary:**

This paper focuses on a way of scaling up GSL via the proposed group discrimination. The authors first analyze the actual reason that makes DGL work and then naturally use one scalar to summarize each node for discrimination in the MI max/min process compared to the usage of a d-dimensional vector in DGI. In summary, the proposed method is somewhat interesting and makes sense even though it is straightforward. However, the quality of this paper is still below the acceptance bar for NeurIPS due to its unclear contribution to the graph domain experts and the lack of theoretical analysis of the proposed method.

**Questions:**

How is Theorem 1 related to the GCL specifically? It seems that it is just an oversmoothing issue in GNN, not caused by GCL.

Why does the accuracy not drop when using group discrimination? How can we ensure this from a theoretical perspective? Since we only use one scalar now compared to the DGI, there should be some information loss.

How is the proposed method specific to the graph domain? It seems that the idea of group discrimination can be easily adapted to the image domain. If so, what is the performance in terms of efficiency and accuracy?

**Limitations:**

The authors are suggested to elaborate more on the motivation part and the relationship with the oversmoothing issue in GNN.

The authors are suggested to provide a theoretical guarantee of the accuracy of the proposed method.

**Strengths And Weaknesses:**

Strengths

The research question is significant. GSL is a hot term nowadays in the GNN domain. The demand for an efficient GSL method is increasing. This work fills this gap.

The logic of this work is quite clear. The authors first analyze why DGI works and how we can simplify it accordingly for efficiency. Then the term "group discrimination" is naturally introduced, and how it can be plugged into the paradigm of GSL is clearly elaborated.

The experiments are extensive and include several well-known large-scale ogbn datasets. The improvement in terms of efficiency is quite tremendous.



Weaknesses

The most critical issue of this work is that the proposed method is not specific to the graph domain. In other words, the proposed method seems to work perfectly even if the input is not the graph, say images. We can also apply the same idea of group discrimination for self-supervised learning on the image classification task. If the proposed method is motivated by the image domain or the general large-scale self-supervised learning, the authors should conduct experiments on the image data as well, as the proposed method is not domain-specific to graph data.

The motivation is unclear. The analysis of the DGI method is seemingly sound, but the mentioned phenomenon has nothing to do with graph contrastive learning. In fact, it is the over-smoothing issue involving GNN itself. Theorem 1 is also well-known[1]. The authors at least add some discussion on the relationship between oversmoothing issue in GNN and the graph contrastive learning. Otherwise, the motivation does not hold. It is not a rethinking of a GCL method but rather a reiteration of the well-known fact in GNN. If the motivation originates from the issue of DGI only, how can we ensure that the proposed method also works for other GCL methods except for DGI?

The proposed method is not quite exciting, and the group discrimination cannot be overexaggerated as a new GCL paradigm but more like a trick or technique. In fact, using one scalar to summarize the whole embedding vector is not quite new. For instance, in binarized GNN[2], we also aim to learn a binarized vector to quantize the embedding for scaling up GNN, which shares a similar idea.

Another critical issue is the lack of direct theoretical analysis. Since only one scalar is used for each node in GCL, how to make sure there is no information loss? Why does it perform even better than classic GCL methods? Are there any theoretical guarantees?


[1] Cai, Chen, and Yusu Wang. "A note on over-smoothing for graph neural networks." arXiv preprint arXiv:2006.13318 (2020).

[2] Wang, Hanchen, et al. "Binarized graph neural network." World Wide Web 24.3 (2021): 825-848.

---

> ### Author Response · Authors · 2022-08-02
> **Response to Reviewer 5yox**
>
> We appreciate Reviewer 5yox for diligent work in reviewing our paper. The reviewer may have some misunderstandings about our paper. Therefore, we summarise some important points to clarify below:
>
> * **Our method is graph-specific.**
> *  **Our paper has no relation to oversmoothing.**
> * **The scalar used in group discrimination is a prediction for a binary classification task instead of node representations.**
> * **Group discrimination is distinct from existing graph contrastive learning (GCL) methods** and cannot be regarded as a trick.
>
> We hope our explanation below addresses the reviewer's concerns. Please let us know if there is still any concern.
>
> **1. Regarding our work is not graph-specific and should run image data.** It is worth noting that our method is a graph-specific approach and cannot be directly applied to image data. Group discrimination (GD) strongly relies on topology information. Specifically, we propose a new pretraining task, GD, to pretrain GNNs. The task focus on discriminating between positive node samples generated with correct topology and negative samples with corrupted topology. We have modified Section 2.1 to improve the clarity. As image data do not have prior graph topology, it is not practical to directly adapt our method to images.
>
> **2.Regarding the relationship of our work with oversmoothing.**  Our method has no relation to oversmoothing. Oversmoothing refers to the convergence of embeddings while graph neural networks go deep, i.e., the number of layers increases.
>
> However, for Theorem 1, it is used to explain the inappropriate usage of Sigmoid function on embeddings generated with Xavier-initialised GNN encoder can lead to dramatic information loss. In specific, these embeddings would become approximately a constant vector. This phenomenon occurs only if Sigmoid is applied to generated embeddings regardless of the depth of the graph neural network. For example, in Table 2, the summary vector statistics are obtained with a 1-layer GCN. To better clarify this, we have modified Theorem 1, its proof, and Table 2.
>
> **3. Regarding our motivation.** The motivation of our work is we point out a misconception in some representative graph contrastive learning (GCL) works. They hypothesise their methods are based upon mutual information (MI). For example, in the first GCL work, DGI stated their model learns from MI maximisation between positive node pairs and MI minimisation between negative pairs. However, we have observed a technical defect in DGI, and thus making a new group discrimination method. Surprisingly, GD is very efficient in time and memory consumption compared with GCL methods.
>
> **4.Regarding our method is similar to binarised GNN and is not new.** Group discrimination (GD) is distinct from binarised GNN. In particular, the summarised scalar in GD only serves as a prediction value to guess if the node embedding is generated with correct graph topology. Specifically, the prediction value (i.e., the summarised scalar) is preferred to be 0 for corrupted samples and 1 for nodes with correct topology. Thus, GD is only a binary classification task for model pretraining. However, in binarised GNN, they use a binary vector to be node representation to improve the efficiency and scalability of GNNs.
>
> This misunderstanding occurs maybe because we use ''summarise embedding information to a scalar'' a lot when describing GD. Actually, ''aggregate embedding to obtain a prediction'' is more accurate. To better clarify this point, we have revised the description of GD in paper.
>
> **5.Regarding our method is a trick for GCL methods.** GD is distinct from GCL. GCL relies on building contrastiveness between node pairs to conduct MI minimisation or maximisation, which requires similarity calculation for embeddings. In contrast, GD requires no contrastiveness (i.e., embedding similarity computation) in training. It is only a binary classification task to distinguish nodes generated with correct and corrupted topology. Moreover, our experimental results show a significant improvement in terms of efficiency on many datasets.
>
> **6.Regarding our method would have information loss as we summarise node embeddings to a scalar.** As explained above, the scalar here is only a prediction for the GD task instead of node representations. In downstream tasks, we use the generated embeddings from the trained model to train and test a linear classifier. Thus, our method has no information loss as we are not using scalars as node representations.
>
> **7.Regarding why our method outperforms DGI.** In section 2.1, we found out that DGI is, in fact, doing group discrimination and thus can be simplified with GD. As GGD is built upon the simplified DGI, our performance should be at least on par with DGI performance. Moreover, we add additional mechanisms, such as global information injection to improve its performance. An ablation study to evaluate the effectiveness of these mechanisms is in Appendix A.10.

---

> > ### Comment · Reviewer_5yox · 2022-08-03
> > **Feedback**
> >
> > Thanks for the detailed rebuttal, and some of my misunderstandings have been addressed. However, there are still some major issues to be addressed directly.
> >
> > For point 1, it is obvious that the proposed method is NOT graph-specific. We can easily apply the idea of GD to the image domain. For example, we can simply change the input in Figure 3 to images and use some image augmentation and corruption methods accordingly. The encoder would be CNN, then. The reviewer cannot see any difficulties in doing so.
> >
> > For point 2, the revised theorem 1 even brings more questions. How does it have anything related to graph contrastive learning? Even if it is true, it is an issue related to GCN itself and has nothing to do with the DGI model directly. The sigmoid function is NOT the key contribution of the DGI model and is just an implementation choice in DGI. Therefore, the motivation of this work is questionable to the reviewer. Also, even if Theorem 1 is true, the reviewer does not see any practical meanings behind it. The feature dimension $D$ is fixed in reality. So what are the insights of "when $D$ increases"? And can you give a more quantitative statement on how large $D$ will make this issue happen? This should be fairly easy, according to your proof. A more formal theorem is expected. After checking the proof, it does not even use anything related to contrastive learning. It only involves with GNN model itself with the activation functions and initialization.
> >
> > Again, the paper still lacks a direct theoretical analysis of the proposed GD method. How this proposed method improves DGI, and in which aspects? More theorems are necessary to make it convincing.

---

> > > ### Author Response · Authors · 2022-08-05
> > > **Thanks for the prompt feedback! Response to the Feedback (Part 1).**
> > >
> > > Thank you for providing feedback to us promptly! The response to the feedback is shown below. We hope our response addresses your concerns.
> > >
> > > **For point 1, it is obvious that the proposed method is NOT graph-specific. We can easily apply the idea of GD to the image domain. For example, we can simply change the input in Figure 3 to images and use some image augmentation and corruption methods accordingly. The encoder would be CNN, then. The reviewer cannot see any difficulties in doing so.**
> > >
> > > **Response:**     Thanks to the reviewer for pointing out the potential of our method in dealing with data outside the graph domain (e.g., images).
> > >
> > > The primary focus of our paper is on graph representation learning. By thoroughly examining the seminar work Deep Graph Infomax (DGI), we identified a pitfall in DGI and thus proposed an efficient method for graph self-supervised learning. The experimental results have demonstrated the significant improvement of our methods over SOTA on various datasets. While our method may have the potential to deal with image data, we believe this should be a separate study. This is because every method, when adapted to another domain, should be taken into consideration of the domain-specific context. A naive and straightforward extension may not end up with good performance. For instance, the Transformer was originally proposed for handling text data, and it was only adapted to the vision domain (Vision Transformer) several years later.
> > >
> > > Furthermore, for extending our method to images, we may have to deal with several challenges below.
> > >
> > > The first issue is the data structure difference between graphs and images. Particularly, in our work, we focus on node classification tasks. Nodes are essential elements in graphs, while for images if we considered them as grids, the according element would be pixels. However, in the image domain, they focus more on image classification instead of the elements in it.
> > >
> > > The second issue is negative sampling. Most existing contrastive learning methods in images use the random negative sampling approach. Specifically, they randomly choose some images from a batch or consider the whole batch as negative samples. However, in group discrimination, we need to have two groups of samples. All original or augmented samples would be considered as in the same group. In graphs, we can easily corrupt the graph structure to generate negative node samples, whereas, for images, we need to design a new negative sampling approach (i.e., the corruption). If we use image augmentation, e.g., colour distortion and rotation, as corruption for images, the generated samples would still be positive samples in the context of group discrimination.
> > >
> > > The third issue is the effectiveness of GD for images. As GD is directly derived from DGI, we know its performance is at least on par with DGI and would guarantee to be effective on graphs. However, it is unsure if our proposed method would work on image data. This should be further investigated in a separate study.
> > >
> > > To summarise, we thank the reviewer for pointing out the great potential of our method. However, adding this part will distract the focus of our paper, and we expect that follow-up works will deal with this problem. We hope our response addresses your concern.
> > >
> > > **For point 2, the revised theorem 1 even brings more questions. How does it have anything related to graph contrastive learning? Even if it is true, it is an issue related to GCN itself and has nothing to do with the DGI model directly.**
> > >
> > > **Response:** We first explain the relationship between Theorem 1 and DGI. Theorem 1 is used to theoretically explain the technical defect we found in the official implementation of DGI. This implementation is used to produce all the reported results in the DGI paper. This technical defect makes the summary vector in DGI become an approximately constant vector and thus has nearly no information. However, from Figure 2 in the paper, we can see the summary vector is the key to the contrastive learning scheme proposed in DGI. DGI stated the success of their scheme is based on the mutual information interaction between the summary vector and node samples. Specifically, they would maximise the similarity between node embeddings and the summary vector. If the summary vector is only a constant vector with no useful information, then it is meaningless to build the contrastiveness between the summary vector and node samples as there won't be mutual information between them. However, DGI is still successful even when the summary vector is meaningless. This leads us to explore what truly leads to the success of DGI, and then we find out,  the first graph contrastive learning work, DGI is actually doing group discrimination instead of graph contrastive learning.

---

> > > > ### Comment · Reviewer_5yox · 2022-08-08
> > > > **Critical flaws in the proof, making the motivation really questionable**
> > > >
> > > > Thanks for the response, and the reviewer really appreciates the efforts. However, Theorem 1 is still far from convincing and contains some obvious errors after checking the proof. Theorem 2 is also trivial, and a more proper and direct analysis of the error rate (see Lemma 1 in the DGI paper) is expected.
> > > >
> > > > In theorem 1, there is still no direct connection with the DGI model itself, and it contains some obvious errors in the proof, making the motivation of the whole paper weak and questionable. Eq.(6) is even wrong since the normalized adjacency matrix should be $\hat{D}^{-\frac{1}{2}} A \hat{D}^{-\frac{1}{2}}$, instead of $\hat{D}^{\frac{1}{2}} A \hat{D}^{\frac{1}{2}}$. Please check the GCN paper, and it is a well-known fact. Even if this is a typo, in lines 544-545, it says, "As multiplying the normalized adjacency matrix will not change the output data range, the output of $\hat{D}^{\frac{1}{2}} A \hat{D}^{\frac{1}{2}}Z$ is still within the range $[0,1]$". This is clearly wrong. First, the output is a matrix, and it should say the entity in this matrix is within the range. Second, there is no guarantee that each row of the normalized adjacency matrix $\hat{D}^{-\frac{1}{2}} A \hat{D}^{-\frac{1}{2}}$ or $\hat{D}^{\frac{1}{2}} A \hat{D}^{\frac{1}{2}}$ sums to 1. Besides, the sum of the column of the normalized $Z$ is also not guaranteed to 1. The reviewer can easily come up with a toy example if you need it. Therefore, the reviewer finds it wrong in this step, making the following proof not trustable. This critical flaw makes this theorem groundless. Consequently, the motivation of this whole paper collapses in some sense.
> > > >
> > > > Besides, this paper lacks a formal analysis of the time complexity of the proposed method. Since the reduced time complexity is claimed to be the main contribution, more theorems on this aspect is also expected. If the time complexity is not the main contribution from the latest manuscript, then more theoretical analysis on its connection with DGI must be added. Why the proposed method performs better than DGI even after the simplification? The reviewer may lower the score if the above-mentioned issues are not directly addressed.

---

> > > > > ### Author Response · Authors · 2022-08-09
> > > > > **Thanks for the Feedback! Response to Reviewer 5yox.**
> > > > >
> > > > > We appreciate Reviewer 5yox for providing the feedback. The response to the feedback is shown below. We hope our response addresses your concerns.
> > > > >
> > > > > **Regarding Theorem 1.**
> > > > >
> > > > > **Response:** Thank you for pointing this out. We have fixed the typo and updated the proof in the Appendix to reflect this. We define $\hat{\textbf{D}}^{-\frac{1}{2}}\hat{\textbf{A}}\hat{\textbf{D}}^{-\frac{1}{2}}$
> > > > > as $\textbf{A}_{norm}$.
> > > > >
> > > > > Then, we multiply $\textbf{A}_{norm}$ with $\textbf{Z}$ to conduct graph convolution, we consider the output matrix as $\textbf{V}$.  For $\textbf{V}_i$, the $i$-th row of $\textbf{V}$, the data range of elements in the vector is within $[0, \sqrt{\frac{d_i}{min(d_j)}}]$, where $d_i$ and $d_j$ are the node degrees for node $i$ and its neighbour $j$, respectively. $min(d_j)$ means the minimum degree in neighbouring nodes of node $i$. It is worth noting that though the data range differs from the previous $[0, 1]$, Theorem 1 still holds. This is because for a node $i$, $\sqrt{\frac{d_i}{min(d_j)}}$ is a constant.
> > > > >
> > > > > **Regarding Theoretical Connection to DGI.**
> > > > >
> > > > > **Response:** We have provided a theoretical analysis of our proposed method to build the connection with DGI. The theoretical analysis is added in Appendix A.2.2.
> > > > >
> > > > > In summary, we first analyse Lemma 1 in DGI. Based on our analysis in Section 2.1, we assume the summary vector in DGI, $s$, is a constant summary vector $\epsilon I$, where $\epsilon$ is the constant in $s$. In addition, we assume $\epsilon$ in $s$ is independent from the distribution of node embeddings $\textbf{H}$, $p(\textbf{H})$ (e.g., in Table 2, the constant value remain unchanged across different datasets). Then, we derive that the error rate for the optimal classifier between the joint distribution of $p(\textbf{H}, s)$ and the product of marginals $p(\textbf{H})p(s)$ is $Er^* = \frac{1}{2}$. Here we can see $Er$ becomes a constant. In addition, the mutual information (MI) between $\textbf{H}$ and $s$ is equal to 0 as their distribution is independent. Thus, under the assumption, we find out the contradiction between Theorem 3 in our paper (i.e., Theorem 1 in DGI) and our findings. Thus, in this case, minimising the classification error in the discriminator will not maximise the MI between $\textbf{H}$ and $s$.
> > > > >
> > > > > Finally, instead of MI maximisation,  we find out DGI with a constant summary vector is doing the same thing as group discrimination without the summary vector (i.e., increasing the Jensen-Shannon divergence between the distribution of positive and negative embeddings). This motivates us to get rid of the summary vector $s$ in DGI loss. Removing the summary vector $s$ is beneficial to the computation efficiency, this is because it eases the burden of gradient computation, e.g., to compute the gradient for $s$, we need to store and use all the parameters in the model to conduct backward propagation. However, we do not need the summary vector in group discrimination and only aggregate node embeddings to obtain predictions.

---

> > > > > > ### Comment · Reviewer_5yox · 2022-08-10
> > > > > > **Thanks for the efforts and hopes the remaining concerns will be fixed later**
> > > > > >
> > > > > > Thanks for the updated feedback. The reviewer highly appreciates the authors' efforts and is deeply moved by the authors' attitude despite the fact that the proof still has some major flaws. Therefore, the reviewer updates the score mainly based on the authors' attitude and hopes these obvious technical concerns or flaws will be addressed in the later version.
> > > > > >
> > > > > > After a simple check, in lines 556-557, the authors claim that "we can easily derive that the output data range of the multiplication of $V_i$ with $W$ is in the range $[-\frac{q}{\sqrt{D}}, \frac{q}{\sqrt{D}}]$" If we follow the idea of analyzing $A_{norm}Z$, and assume each element in $V_i$ is upper-bounded by $q$ while each element in $Z$ is upper-bounded by $\frac{1}{\sqrt{D}}$, then we should get the upper-bound of $V_i W$ as $q \sqrt{D}$. Note that this is the matrix multiplication, and the dimension of $V_i$ is $\mathbb{R}^{D}$ and the column of $W$ is also $D$-dimensional. Therefore, this fact will change the results of the following proof, and the convergence results will be changed as well. However, this upper bound is loose, and we may use some probability tools to get more strong bound so that we may achieve this claimed result. This leaves the authors to fix this gap.
> > > > > >
> > > > > > For the added theorems in Appendix 2.2, the reviewer still finds them not directly connected to the proposed GD method, and most of them are similar to the ones in the DGI paper.

---

> > > ### Author Response · Authors · 2022-08-05
> > > **Thanks for the prompt feedback! Response to the Feedback (Part 2).**
> > >
> > > **Regarding a quantitative statement for $D$ that will make the issue happen.**
> > >
> > > **Response:** The feature matrix $D$ is fixed, but $D$ is quite large in all datasets (i.e., Cora, CiteSeer, and PubMed) DGI conduct experiments on. Specifically, $D$ for Cora, CiteSeer, and PubMed are 1433, 3703, and 500, respectively. $D$ in these datasets are big enough to narrow the data range of the summary vector to a tiny interval. To prove this point, taking non-linear activation PReLU (i.e., the one used in DGI) as an example, we present the change of lower and upper bound of the summary vector when $D$ increases as follows:
> > >
> > > | $D$ | 1 | 10 | 100 | 1000 | 10000 |
> > > |---|---|---|---|---|---|
> > > | Lower | 0.5000 | 0.5000 | 0.5000 | 0.5000 | 0.5000 |
> > > | Upper | 0.5621 | 0.5197 | 0.5062 | 0.5019 | 0.5006 |
> > > | Range | 0.0621 | 0.0197 | 0.0062 | 0.0019 | 0.0006 |
> > >
> > > From the table above, we can see when $D$ is over 100, the data range is already very small. Instead of when ''$D$ increases'', ''when $D$ is large, the summary vector would become approximately a summary vector'' would be more appropriate. We have modified the manuscript to improve the clarity.
> > >
> > > **Theoretical analysis and how this proposed method improves DGI, and in which aspects?**
> > >
> > > **Response:** We have added a theoretical analysis of the proposed GD method.  In summary, we prove that optimising the group discrimination loss is similar to maximising the Jensen-Shannon divergence between the distribution of positive and negative embeddings.  Thus, group discrimination can help the model avoid making 'mistakes' (i.e., bias the encoder towards avoiding mistaken samples).  To explain this, we introduce a new theorem shown below:
> > >
> > > **Theorem 2** *Given a graph $\mathcal{G}$, a corrupted graph $\tilde{\mathcal{G}}$, and a encoding network $f_{\theta}()$, we consider the distribution of positive embeddings $f_{\theta}(\mathcal{G})$ as $P_{pos}$ and negative embeddings $f_{\theta}(\tilde{\mathcal{G}})$ as $P_{neg}$. Optimising the group discrimination loss is equivalent to maximising the Jensen-Shannon divergence between $P_{pos}$ and $P_{neg}$.*
> > >
> > > Baed on the Theorem, as $P_{pos}$ would be separated from $P_{neg}$, the positive node embeddings $f_{\theta}(\mathcal{G})$ become separated from $f_{\theta}(\tilde{\mathcal{G}})$, and thus improves the quality of node embeddings generated with $\mathcal{G}$. An intuitive explanation and detailed proof for the theorem have been added to Appendix A.2.1.
> > >
> > > For how the proposed method improves DGI, firstly, GGD boosts the efficiency of DGI in training time consumption. Remarkably, GGD is orders of magnitude (10,000+) faster than graph contrastive learning baselines on the Ogbn-arxiv dataset. Unlike graph contrastive learning, instead of conducting similarity computation between node embeddings, GD only conducts a simple binary classification task.
> > >
> > > In addition, as explained in Section 2.1, we have found out that DGI is, in fact, a group discrimination method. Thus, we can simplify DGI with GD loss. Then, we further improve simplified DGI by adding three components upon it, including augmentation, the projector and the global information injection operation. We have conducted an ablation study to evaluate the effectiveness of different components in GGD (upon the simplified DGI). Specifically, we evaluate three variants of GGD, including $GGD_{w/o\ aug}$, $GGD_{w/o\ proj}$, and $GGD_{w/o\ power}$, which represent GGD without augmentation, the projector and the global information injection process, respectively. The experiment results on five small to medium size datasets are presented in the table below, where we can see without any mentioned component, the performance of GGD degrades. This validates the effectiveness of these components.
> > >
> > > | Method | Cora | CiteSeer | PubMed | Comp | Photo |
> > > |---|---|---|---|---|---|
> > > | $GGD_{w/o\ aug}$ | 83.6$\pm{0.3}$ | 72.4$\pm{0.4}$ | 81.2$\pm{0.2}$ | 89.6$\pm{0.4}$ | 92.2$\pm{0.5}$ |
> > > | $GGD_{w/o\ proj}$ | 82.9$\pm{0.4}$ | 71.0$\pm{0.4}$ | 79.6$\pm{0.1}$ | 89.4$\pm{0.5}$ | 91.6$\pm{0.4}$ |
> > > | $GGD_{w/o\ power}$ | 83.0$\pm{0.5}$ | 72.5$\pm{0.4}$ | 80.1$\pm{0.4}$ | 89.9$\pm{0.6}$ | 92.3$\pm{0.4}$ |
> > > | $GGD$ | 83.9$\pm{0.4}$ | 73.0$\pm{0.6}$ | 81.3$\pm{0.8}$ | 90.1$\pm{0.9}$ | 92.5$\pm{0.6}$ |
> > >
> > > Then, we explain why the added three components are effective. For the augmentation, we conjecture it can increase the difficulty of the group discrimination task as both the input graph structure and the feature matrix would change in every training iteration. This can force the model to lessen the dependence on the fixed pattern in a monotonous graph. For the projector, it is an additional multilayer perceptron to increase the depth of the model. Finally, for the global information injection, we are inspired by MVGRL, a representative GCL method, as explained in Section 3.2. The global information injection can reinforce the generated embeddings with additional global information.

---

> > > ### Author Response · Authors · 2022-08-09
> > > **Latest comments for Reviewer 5yox**
> > >
> > > Dear Reviewer,
> > >
> > > We have provided a detailed response to your questions. We haven't heard back from you. We hope our response helps clarify your remaining concerns and reconsider your perception and rating of our work.
> > >
> > >
> > > Best,
> > > Authors

---

> > > > ### Comment · Reviewer_5yox · 2022-08-10
> > > > **Reply**
> > > >
> > > > Dear authors,
> > > >
> > > > The reviewer highly appreciates the efforts you made and has updated the score.  Thanks!
> > > >
> > > > Best,

---

### Official Review · Reviewer_92EW · 2022-07-11

**Rating:** 6
**Confidence:** 4
**Soundness:** 3 good
**Presentation:** 2 fair
**Contribution:** 3 good

**Summary:**

I enjoyed reading this unusual paper. This paper proposes a new graph pretraining pre-text task -- group discrimination -- with low computational cost and fast convergence speed. The idea is simple -- pre-train a GNN to discriminate the node embeddings of original graphs and the nodes from corrupted graphs with randomly shuffled nodes. Besides group discrimination, the method also uses a diffusion operator to enhance the node embeddings. This method achieves state-of-the-art or comparable performances on node classification benchmarks.

The method is inspired by an interesting observation in the previous DGI and MVGRL works. By analysis, the authors show that the supposedly contrastive objective in previous works is actually doing group discrimination. The authors then simplify the existing methods into this group discrimination pre-text task. The new pre-text task shows lower complexity and faster convergence speed.


**Questions:**

Q1: graph diffusion models like SGC have shown promising feature extraction capability without any pre-training. GGD shares the similar spirit of using the power of adjacency matrix to obtain global embedding. How does GGD perform without the diffusion operator? How does the baseline perform if the diffusion operator is included?

Q2: how does GraphCL perform in the given benchmarks?


**Strengths And Weaknesses:**

The method is surprisingly simple but effective. It is also new and original in self-supervised learning for graphs. In experiments, the method shows significant improvement than baselines in terms of complexity and performance.

However, I find some incorrect claims of complexity and missing ablation study while reading. These weaknesses can hurt the credibility of the proposed method.



Weaknesses:

- The ablation study for the graph diffusion module is missing. Doe the group discrimination perform as good when the graph diffusion model is removed?
- Missing baseline: GraphCL. In Line218, the model uses the augmentation from GraphCL. How does GraphCL perform in the presented benchmarks?
- Some claims about the complexity are incorrect. I didn't verify all the complexity claims, but here are examples:
  - The complexity of the loss function for one node is actually O(d) because of the sum operator for node embeddings in Line173. The authors claim the complexity if O(1) throughout the paper. Despite the O(1) complexity, the theoretical improvement is weak. The time complexity of training GNNs is dominated by the GNN forward of at least O(EDL) (E is the edge number). Compared to GNN forward, the improvement on the time complexity of loss function is negligible.
  - The complexity of BarlowTwins loss is actually O(ND^2) for a batch of N nodes. This complexity is due to calculating the cross-correlation matrix using two matrices of shapes (D, N) and (N, D). However, the authors claim the complexity for one node is O(D) in Line50-51.
  - The complexity of GCN is actually O(LMD + LND^2) rather than the claimed value in Line648-649. See Cluster-GCN for the analysis of GCN's complexity.

- Some important experimental details are missing.
  - It is unclear if the authors freeze the encoder while linear evaluation or do they fine-tune the entire model?
  - Is the learning rate in Table15 used for pre-training or fine-tuning?



Typos:

- Line123: are served --> serve
- Line142: Theorem and detailed --> Theorem

---

> ### Author Response · Authors · 2022-08-02
> **Response to Reviewer 92EW**
>
> We appreciate Reviewer 92EW for the thorough review and insightful feedback. We hope our responses address all weaknesses and questions. Please let us know if there is any concern. We have modified our manuscript to improve the clarity, and changes have been highlighted.
>
> **1. Regarding ablation study for graph diffusion.** To evaluate the effect of the global information injection operation, we have conducted an ablation study, whose results have been added to Appendix A.10. From the results, the performance of GGD degrades on all datasets without this operation. However, compared with self-supervised baselines (as shown in Table 5 of the paper), it still achieves the highest performance in 4 out of 5 datasets. This result validates the effectiveness of GGD even without global information injection.
>
> **2.Regarding the baseline GraphCL** We have conducted experiments for GraphCL in terms of their performance, training time and memory consumption. The results are added in Section 5.1.
>
> **3.Regarding claims for complexity.**
>
> * **The author claims $O(1)$ for the loss function** It is worth noting that the summation is conducted prior to the loss computation. In practice, the summation is conducted in the ''aggregation'' phase, as shown in Figure 3, to obtain the prediction value (preferably close to 0 for positive or 1 for negative samples) for the binary group discrimination (GD) task. Moreover, summation is only one way to do aggregation. As shown in Appendix A.3, we can do other aggregation, e.g., pooling, to obtain the prediction.
>
> Thus, we claimed the complexity of the loss function for one node is $O(1)$ as the loss function is only a BCE loss as below:
>
> $\mathcal{L}_{BCE}(i) = y_i\log h_i + (1 - y_i)\log(1 - h_i)),$
>
> where $ y_i \in \mathcal{R}^{1 \times 1}$ means the indicator for a node sample $i$ (i.e., if node sample $i$ is corrupted, $ y_i$ is 0, otherwise it is 1), and $ h_i \in \mathcal{R}^{1 \times 1}$ is the prediction value. As the computation only requires the multiplication of two scalars. We consider its complexity as $O(1)$. We have modified Section 2.1 to clarify this.
>
> **In addition, our method aims to reduce time complexity for graph contrastive learning (GCL) methods instead of reducing the complexity for GNN encoders.** Existing GCL methods are inefficient as they adopt complicated contrastive losses upon GNN encoders, which require similarity computation between vectors. Instead of calculating similarity, the GD paradigm is only a binary classification task to pre-train the GNN model. Thus, it is simpler than existing GCL methods. To better clarify this, we present the comparison of time complexity of 4 representative contrastive losses and GD loss for processing the whole graph as below:
>
> | Loss | InfoNCE | JSD | BGRL | GBT | GGD |
> |---|---|---|---|---|---|
> | Complexity | $O(N^2D)$ | $O(ND)$ | $O(ND)$ | $O(ND^2)$ | $O(N)$ |
>
> * **The complexity of BarlowTwins loss is $O(ND^2)$, while authors claim it is $O(D)$ for one node.** Thank you for pointing this out. We initially want to explain here that the complexity of loss computation for one embedding dimension in BarlowTwins is $O(ND)$. This is because, for one embedding dimension, BarlowTwins requires cosine similarity computation for $D$ dimensions. Each computation requires $O(N)$ for two $1 \times N$ embedding dimension vectors. In total, the time complexity for one embedding dimension is $O(ND)$. We have rectified according part mentioned GBT complexity and Appendix A.2.
>
> * **The complexity of GCN is actually $O(LMD + LND^2)$** The complexity of GCN conflicts in many studies. For example, we adopt the time complexity of GCN from a popular survey for Graph Neural Networks with thousands of citations [1]. Table 3 of their paper claimed that the time complexity for GCN is $O(M)$, where $M$ represents the number of edges in a graph. In our paper, we consider the time complexity of the encoding part as $O(N + M)$ because we need to corrupt the feature matrix $\textbf{X}$, which requires $O(N)$.
>
> Another example is BGRL. In their computational complexity analysis, they claimed that the most popular GNNs, including GCN, share the same complexity $O(N + M)$ in time and spaces. These proposed complexities are different from the ones in cluster-GCN. We conjecture this is because they use a simplified version of time complexity for GCN, which only considers the message passing operation.
>
> To be more specific for the complexity analysis, we decided to adopt the Cluster-GCN complexity, which is detailed. Therefore, we have revised Appendix A.5 to reflect this change.
>
> **4. Regarding if freeze the encoder while doing linear evaluation/what the learning rate in Table 15 means.**
>
> Yes, we freeze the encoder. The learning rate is for pre-training. We have modified the manuscript to clarify.
>
> [1] A comprehensive survey on graph neural networks. TNNLS 2020.
>
> [2] Large-scale representation learning on graphs via bootstrapping. ICLR 2022.

---

> > ### Comment · Reviewer_92EW · 2022-08-06
> > **Comment on the Authors' Rebuttal**
> >
> > Thanks for the detailed rebuttal. I appreciate the authors' efforts in presenting a better ablation study and running more baselines. Some of my concerns are addressed. However, I would like to point out that the following issues remain to be solved.
> >
> > * **Theorem1:** Theorem 1 is used to demonstrate the defect of DGI that the summary vectors become a constant vector with the sigmoid activation. However, the theorem is invalid since the result is limited to a randomly initialized model.
> >
> > - **Complexity Claims:** While I am persuaded that the algorithm converges faster, the complexity claims in the paper are still problematic. The authors are encouraged to present a more detailed and accurate complexity analysis of related works if they are to claim complexity improvement. **I tend to reject if the following issues will not be fixed.**
> >   - The claimed O(1) complexity is misleading and should be corrected. It is true only because the aggregation operation is not counted for loss computation.
> >   - The claimed complexity improvement in loss function is trivial if the overall complexity is dominated by the GNN encoder, which is valid for a GCN encoder. Besides, there is no improvement in complexity compared to JSD [2] and BGRL [3] if we count the aggregation operation in the loss computation.
> > - **Motivation & Explanation:** Fixing a bug from previous studies is not a convincing explanation for a good empirical performance. The authors are encouraged to develop an explanation (intuitively or theoretically) for the group discrimination loss function. As the authors mentioned in the paper, DGI and MVGRL are not actually contrasting representations, then the explanation for contrastive learning cannot be applied here. Owning to the discrimination loss between real and fake samples, the explanation for Generative Adversarial Networks [1] might be a plausible direction.
> >
> > Due to these issues, I reserve the previous rating.
> >
> >
> >
> > **Reference:**
> >
> > [1] Goodfellow et al. Generative adversarial nets. In NIPS 2014.
> >
> > [2] Velicˇkovic ́ et al. DEEP GRAPH INFOMAX. In ICLR 2019.
> >
> > [3] Thakoor et al. Large-scale representation learning on graphs via bootstrapping. In ICLR 2022.

---

> > > ### Author Response · Authors · 2022-08-08
> > > **Thanks for the Feedback! Response to Reviewer 92EW.**
> > >
> > > We appreciate Reviewer 92EW for the informative feedback! In short, we have removed all complexity improvement claims and provided a theoretical analysis of group discrimination. The response to the feedback is presented below. We hope our response addresses your concerns.
> > >
> > > **Regarding Theorem 1.**
> > >
> > > **Response:** Based on the updated Table 2, we can see the constant in the summary vector remain unchanged, and the information loss still occurs even if the GNN encoder is trained. Thus, we conjecture the training process won't affect the constant value much in the summary vector of DGI. Therefore, to some extent, Theorem 1 can still explain why the value in the summary vector would converge to a certain value (e.g., 0.5 or 0.62). We have modified Theorem 1 to state it is valid at the initialisation stage.
> > >
> > > **Regarding Complexity Claims.**
> > >
> > > **Response:**  To avoid the controversial claim, We have removed all parts regarding the $O(1)$ complexity and all claims regarding complexity improvement in the paper. We hope this address the reviewer's concern.
> > >
> > > **Regarding Motivation and Explanation for Group Discrimination.**
> > >
> > > **Response:** Thank you for your advice! We are inspired by the theoretical proof for GAN and conducted a theoretical analysis for the group discrimination loss. In summary, optimising the group discrimination loss is equivalent to maximising the Jensen-Shannon divergence between the distribution of positive and negative embeddings. In other words, group discrimination is actually learning to avoid making `mistakes' (i.e., bias the encoder towards avoiding mistaken samples). To explain this, we introduce a new theorem shown below:
> > >
> > > **Theorem 2** *Given a graph $\mathcal{G}$, a corrupted graph $\tilde{\mathcal{G}}$, and a encoding network $f_{\theta}()$, we consider the distribution of positive embeddings $f_{\theta}(\mathcal{G})$ as $P_{pos}$ and negative embeddings $f_{\theta}(\tilde{\mathcal{G}})$ as $P_{neg}$. Optimising the group discrimination loss is equivalent to maximising the Jensen-Shannon divergence between $P_{pos}$ and $P_{neg}$.*
> > >
> > > As $P_{pos}$ would be separated from $P_{neg}$, the positive node embeddings $f_{\theta}(\mathcal{G})$ become separated from $f_{\theta}(\tilde{\mathcal{G}})$ by maximising the Jensen-Shannon divergence.  Thus, the quality of the positive node embeddings can be improved by avoiding being similar to out-of-distribution samples. An intuitive explanation and detailed proof for the theorem have been added to Appendix A.2.1.

---

> > > > ### Comment · Reviewer_92EW · 2022-08-09
> > > > **Feedback on rebuttal**
> > > >
> > > > The authors' response have addressed my concerns in the complexity claims. I have raised my rating from 5 to 6.

---

### Official Review · Reviewer_2iwV · 2022-07-25

**Rating:** 7
**Confidence:** 4
**Soundness:** 3 good
**Presentation:** 3 good
**Contribution:** 3 good

**Summary:**

The paper presents a novel approach to graph representation learning that learns informative node embeddings by projecting the embeddings obtained from the original graph using a GCN into scalars, and the embeddings obtained from a corrupted version of the graph using the same GCN into scalars, and optimizing the parameters of the model so that the embeddings of the original graph are all distinguishable from the embeddings of the corrupted version of the graph using a logistic classifier. The proposed method, Graph Group Discrimination, is shown to use much less memory, converge much faster, be more scalable, and to usually be more accurate than prior models when applied to node classification tasks.

**Questions:**

- Explain how the graph representations learned by GGD and competing methods are actually used to compute the reported node classifications. E.g. are the same architectures used?

- State that the Figure 5 mentioned in Section 3 is in the appendix.

- The current description in Section 3 is not sufficient. Give an unambiguous loss function, and algorithm listing for minimizing this loss. Explain what the projector and aggregation steps are, and what architectures are used to parameterize/accomplish them, in the same way that it is explained a GCN is used for the encoder.

- It is mentioned that global information can be included in the GGD method in Section 3.2, but it is not clear whether this is used in any of the experiments from the main body of the paper. This should be clarified (i.e. no, none used global infromation), and the reader pointed to the Appendix for experiments using global information.

- I had questions about the data sets and specific architectures and hyperparameters used in the experiments. The former (data sets) were addressed in the Appendix; this should be pointed out in the paper. However, the experiments are not well specified because I do not see what the architectures or hyperparameters were used; please give this information in the Appendix as well. Also, what code did the authors use for the two baselines BGRL and GBT: did they reimplement them, or use existing code? This should be mentioned in section 5.1.

Overall, I thought the paper's contribution is significant, but the presentation needs work to address the above issues before I can give it a solid accept.

**Limitations:**

Yes.

**Strengths And Weaknesses:**

Strengths:
- The approach is novel, fast, and more scalable than previous graph contrastive learning methods.
- The method leads to embeddings that are accurate for node classification.
- The overview of GCL methods is informative and positions the contribution clearly.

Weaknesses:
- It is not stated how the learned graph representations are used to compute the node classifications show in the results.
- No explicit algorithm is stated, so some steps are ambiguous (e.g., it's never made clear how a "projection" and "aggregation" are in Figure 3; it's also not clear from that figure that the augmentation and corruption change between training iterations)
- I don't see why the claim in Theorem 1 about the GCN encoder leading to constant summaries is relevant: this result assumes that the encoder is not being trained, but it is, so the result does not accurately reflect what is going on in DGI. The empirical results of the author in
Table 2 aren't explained by Theorem 1.

---

> ### Author Response · Authors · 2022-08-02
> **Response to Reviewer 2iwv**
>
> We appreciate Reviewer 2iwv for the perception of our contributions and thank the reviewer for the insightful feedback. We hope our responses address all the reviewer proposed weaknesses and questions. Please let us know if there is still any concern. Changes to the manuscript have been highlighted. The detailed responses are shown below:
>
> **1. Regarding how the learned graph representations are used to compute the node classifications shown in the results.**
>
> **Response:** To conduct the downstream node classification task, we follow the general practice of self-supervised graph neural network methods. In specific, after pre-training, we input the graph dataset to the trained model to obtain trained embeddings. These embeddings are then used to train and test a simple linear (i.e., logistic regression) classifier. To clarify this point, we have added the description of this process in section 3.2.
>
> **2. Regarding the explicit algorithm, ambiguous steps and unclear description of ''Projection'' and ''Aggregation'' in Figure 3.**
>
> **Response:** The ''projection" part is only a multilayer perceptron (MLP) network whose number of layers can be adjusted as required. The ''aggregation" part means aggregation techniques including summation, pooling, and linearly aggregation, which can aggregate a node embedding vector into a scalar. To better illustrate these two components, we modify the caption in Figure 3 and section 3.1.
>
> In addition, we have summarised the overall procedure of GGD in Algorithm 1, which is added to Appendix A.9.
>
> **3. Regarding the relevance of Theorem 1 and its connection to Table 2**
>
> **Response:** Sorry for the ambiguity of Theorem 1. This theorem is used to show the inappropriate usage of the Sigmoid function in DGI can cause dramatic information loss in the summary vector because the vector would approximately become a constant vector. We further improved and modified our proof for Theorem 1, which now shows that the summary vector in DGI would contain approximately 0.5 using ReLU/LReLU/PReLU or 0.62 using Sigmoid as non-linear activation for the GNN encoder at the initialisation stage. We have further conducted experiments for the statistics of the summary vector and updated Table 2.
>
> We found it isn't easy to prove Theorem 1 when the GNN encoder is trained. Nonetheless, based on the updated Table 2, we can see the constant in the summary vector remain unchanged, and the information loss still occurs even if the GNN encoder is trained. Thus, we conjecture the training process won't affect the constant value much in the summary vector of DGI.
>
> Theorem 1 and Table 2 are updated in the manuscript and shown below:
>
> **Theorem 1** _Given $\mathcal{G} = (\textbf{X} \in \mathcal{R}^{N\times D}, \textbf{A} \in \mathcal{R}^{N\times N})$, and a GCN encoder $g()$ initialised with Xavier initialisation, we can obtain its embedding $\textbf{H} = g(\mathcal{G})$. Then, when $D$ increases, the value in $\textbf{H}$ converges to 0.5 with ReLU/LReLU/PReLU or 0.62 with Sigmoid or as non-linear activation for $g()$._
>
> **Table 2:** Summary vector statistics on three datasets with different activation functions, including ReLU, LeakyReLU (i.e., LReLU shown below), PReLU, and Sigmoid.
>
> |Activation|Statistics|Cora|CiteSeer|PubMed|
> |------------------|------------|---------|----------|---------|
> ||Mean|0.50|0.50|0.50|
> | ReLU/LReLU/PReLU|Std|1.3e-03|1.0e-04|4.0e-04|
> ||Range|1.4e-03|8.0e-04|1.5e-03|
> ||
> ||Mean|0.62|0.62|0.62|
> | Sigmoid|Std|5.4e-05|2.9e-05|6.6e-05|
> ||Range|3.6e-03|3.0e-03|3.2e-03|
>
> **4. Regarding the usage of global information.**
>
> **Response:** As specified in Algorithm 1, we use the global information injection in the inference phase to strengthen the generated embeddings for downstream tasks. All experiments for GGD in the paper are conducted with the injection of global information. We have added description in section 3.2 to reflect this point. It is worth noting that even without the use of global information, GGD still outperforms self-supervised baselines on 4 out of 5 datasets. The ablation study for the use of global information is added in Appendix A.10.
>
> **5.  Regarding experiment settings, parameter settings and datasets.**
>
> **Response:** We have pointed out the dataset statistics in Appendix A.7 in the first paragraph of section 5. The architectures or hyperparameters we used in the experiment are shown in Table 15 in Appendix A.8. Based on this table, we can see the model architecture and hyperparameters used in experiments. For example, from the table, we can see that the model architecture for training Cora consists of a one-layer GCN encoder and a one-layer multi-layer perceptron (i.e., a linear layer) as the projector. In addition, the hidden dimension and the learning rate are set to be 256 and 1e-3, respectively.
>
> In addition, we use their official implementation in the experiment for the two baselines, BGRL and GBT. We have added an description in section 5.1.

---

> > ### Comment · Reviewer_2iwV · 2022-08-09
> > **Thanks for the changes**
> >
> > Thank you for addressing the issues I raised. I will take a look at the revision to verify then change my decision to accept.

---

### Official Review · Reviewer_o3aj · 2022-07-31

**Rating:** 7
**Confidence:** 4
**Soundness:** 3 good
**Presentation:** 4 excellent
**Contribution:** 4 excellent

**Summary:**

DGI has shown successful performance as the graph contrastive learning method.
This paper points out the technical defect of DGI, that is, the sigmoid function in the loss makes the embedding uniform.
From that observation, the authors suggest to modify the original loss to much more simplified version (i.e., graph discrimination).
They propose GGD based on the graph discrimination loss, and show the experimental results to prove its efficiency.

**Questions:**

- What is the rationale for saying that "GD-based method focuses on the general edge distribution of graphs instead of node-specific information"?
- How are the training time and memory consumption of the supervised GNN models in the small- and medium-scale datasets?
- Please clarify which results are reproduced and which are sourced from other papers in other sections as in Section 5.1, and indicate how many epochs the models are trained in every case.
- I was curious about the behavior in convergence of the models until 300 and 100 epochs in Figure 4.

**Limitations:**

- As the authors mentioned, the group discrimination can depend on augmentation or corruption methods.
- It would be more persuasive if the experiments are conducted on the datasets from more various domains.

**Strengths And Weaknesses:**

**Strengths**
- They improve both the performance and the resource efficiency.
- It is a well-organized paper: easy to catch the idea, and contains detailed but concise explanation.

**Weaknesses**
- It would be better if there are more empirical explanations how GGD achieves the higher performance than DGI, for example, by the ablation study.
- There are minor mistakes: overlapping letters in Table 9, unit error in line 387 (% $\rightarrow$ %p).

---

> ### Author Response · Authors · 2022-08-02
> **Response to Reviewer o3aj**
>
> We appreciate Reviewer o3aj for the perception of our contributions and thank the reviewer for the insightful feedback. The detailed responses are shown below:
>
> **1. Regarding more empirical explanations of how GGD achieves higher performance than DGI, for example, by the ablation study.**
>
> **Response:** We have conducted an ablation study to evaluate the effectiveness of different components in GGD (upon the simplified DGI). Specifically, we evaluate three variants of GGD, including $GGD_{w/o\ aug}$, $GGD_{w/o\ proj}$, and $GGD_{w/o\ power}$, which represent GGD without augmentation, the projector and the global information injection process, respectively. The experiment results on five small to medium size datasets are presented in the table below. From the table, we can see without any mentioned component, the performance of GGD degrades, which validates the effectiveness of these components. We conjecture why $GGD$ outperforms DGI is because of the addition of these components.
>
> | Method | Cora | CiteSeer | PubMed | Comp | Photo |
> |---|---|---|---|---|---|
> | $GGD_{w/o\ aug}$ | 83.6$\pm{0.3}$ | 72.4$\pm{0.4}$ | 81.2$\pm{0.2}$ | 89.6$\pm{0.4}$ | 92.2$\pm{0.5}$ |
> | $GGD_{w/o\ proj}$ | 82.9$\pm{0.4}$ | 71.0$\pm{0.4}$ | 79.6$\pm{0.1}$ | 89.4$\pm{0.5}$ | 91.6$\pm{0.4}$ |
> | $GGD_{w/o\ power}$ | 83.0$\pm{0.5}$ | 72.5$\pm{0.4}$ | 80.1$\pm{0.4}$ | 89.9$\pm{0.6}$ | 92.3$\pm{0.4}$ |
> | $GGD$ | 83.9$\pm{0.4}$ | 73.0$\pm{0.6}$ | 81.3$\pm{0.8}$ | 90.1$\pm{0.9}$ | 92.5$\pm{0.6}$ |
>
> **2. Regarding the rationale for "GD-based methods focuses on general edge distribution of graphs".**
>
> **Response:** We conjecture GD-based methods focus on the general edge distribution of graphs as the group discrimination task discriminates node samples generated with correct graphs or corrupted graphs. To achieve this, we think GD-based methods should have learned the general edge distribution of the graphs. Specifically, when a node sample is fed to the network, the GD-based methods can compare it with the edge distribution it learned. In other words, if the sample is more similar to samples generated with correct edge distribution, the prediction value would be close to 1, and vice versa.
>
> **3. Regarding additional experiments for convergence and supervised GNNs.**
>
> **Response:** We feel sorry that due to the time and resource limitations, we are not able to conduct these experiments in a short time as, in total, they can consume hundreds of hours. Nevertheless, we will try our best to include these experiments in the later version of our paper.
>
> **4. Regarding clarifying the source of results in Section 5.1.**
>
> **Response:**  We have modified Section 5.1 to clarify the source of the results.

---

### Author Response · Authors · 2022-08-02
**General response to all reviewers**

We thank all reviewers very much for the extensive review and valuable feedback on our paper. We appreciate that they found the contribution of our method is significant [Reviewer o3aj, Reviewer 2iwv], surprisingly simple but effective [Reviewer 92EW], well-organised and clear [Reviewer o3aj, Reviewer 5yox]. According to the reviewer's comments, we have added experiments and the algorithm, rectified the theorem and its according proof, and revised the manuscript to improve its clarity and presentation. Changes to the manuscript have been highlighted in the revised version. Again, we appreciate all reviewers for their constructive comments to further strengthen our work.

---

### Meta-Review · Area_Chair_3Ccw · 2022-09-01

**Recommendation:** Accept
**Confidence:** Less certain

**Metareview:**

The paper presents a novel contrastive method of graph representation learning motivated by the observation that previous approaches to graph contrastive learning actually do group discrimination. The proposed method learns graph representations such that the representations of the original graph and a corrupted version are easily distinguishable when projected to a scalar space. The reviewers found the method to be simple and comparable in performance with state-of-the-art methods, while more scalable.

**Award:**

No

---

### Decision · Program_Chairs · 2022-09-14

Accept